ecology

carnivore conservation, deforestation, endemic species, forest fragmentation, multi-scale analysis, species distribution

**Author for correspondence:**
Daniel G. Rocha
e-mail: rochadg.bio@gmail.com

# Wild dogs at stake: deforestation threatens the only Amazon endemic canid, the short-eared dog (*Atelocynus microtis*)

Daniel G. Rocha[1,2], Katia Maria Paschoaletto Micchi de Barros Ferraz[3], Lucas Gonçalves[4,5], Cedric Kai Wei Tan[6], Frederico G. Lemos[7,8], Carolina Ortiz[3], Carlos A. Peres[9], Nuno Negrões[10], André Pinassi Antunes[11,12], Fabio Rohe[13], Mark Abrahams[14], Galo Zapata-Rios[15], Davi Teles[9], Tadeu Oliveira[16], Eduardo M. von Mühlen[17], Eduardo Venticinque[17], Diogo M. Gräbin[2,18], Diego Mosquera B.[19], John Blake[20], Marcela Guimarães Moreira Lima[21], Ricardo Sampaio[22], Alexandre Reis Percequillo[23], Felipe Peters[24], Esteban Payán[25], Luiz Henrique Medeiros Borges[26], Armando Muniz Calouro[27], Whaldener Endo[28,29], Renata Leite Pitman[30], Torbjørn Haugaasen[31], Diego Afonso Silva[32], Fabiano R. de Melo[33], André Luis Botelho de Moura[34], Hugo C. M. Costa[18], Camile Lugarini[35], Ilnaiara Gonçalves de Sousa[36], Samuel Nienow[35], Fernanda Santos[26,37], Ana Cristina Mendes-Oliveiras[38], Wezddy Del Toro-Orozco[2], Ana Rafaela D'Amico[35], Ana Luisa Albernaz[39], André Ravetta[40], Elaine Christina Oliveira do Carmo[41], Emiliano Ramalho[2,24], João Valsecchi[42,43], Anthony

J. Giordano[44,45], Robert Wallace[46], David W. Macdonald[6] and Rahel Sollmann[1]

[1]Department of Wildlife, Fish, and Conservation Biology, University of California – Davis, Davis, CA, USA

[2]Grupo de Pesquisa em Ecologia e Conservação de Felinos na Amazônia, Instituto de Desenvolvimento Sustentável Mamirauá, Tefé, AM, Brazil

[3]Laboratório de Ecologia, Manejo e Conservação de Fauna (LEMaC), Departamento de Ciências Florestais, Escola Superior de Agricultura Luiz de Queiroz, Universidade de São Paulo, Piracicaba, SP, Brazil

[4]Departamento de Biologia, Universidade Federal Rural de Pernambuco, Recife, PE, Brazil

[5]University of Brasilia, Brasilia, DF, Brazil

[6]Wildlife Conservation Research Unit, Department of Zoology, University of Oxford, The Recanati-Kaplan Centre, Tubney House, Tubney, Oxon, England

[7]Departamento de Ciências Biológicas, Unidade Acadêmica Especial de Biotecnologia, Universidade Federal de Catalão, GO, Brazil

[8]Programa de Conservação Mamíferos do Cerrado/PCMC, Araguari, GO, Brazil

[9]School Environmental Sciences, University of East Anglia, Norwich, UK

[10]Bolivian Association for Research and Conservation of the Andean-Amazon Ecosystems-ACEAA, Bolivia

[11]RedeFauna – Rede de Pesquisa em Diversidade, Conservação e Uso da Fauna da Amazônia, Brazil

[12]Instituto Nacional de Pesquisas da Amazônia, Manaus, AM, Brazil

[13]Programa de Pós-graduação em Genética, Conservação e Biologia Evolutiva –GCBEv. Instituto Nacional de Pesquisas da Amazônia (INPA), Manaus, AM, Brazil

[14]Field Conservation and Science Department, Bristol Zoological Society, Bristol, UK

[15]Wildlife Conservation Society – Ecuador Program, Quito, Ecuador

[16]Departamento de Biologia, Universidade Estadual do Maranhão, São Luís, MA, Brazil

[17]Departamento de Ecologia, Universidade Federal do Rio Grande do Norte (UFRN), Natal, RN, Brazil

[18]Programa de Pós-graduação em Ecologia e Conservação da Biodiversidade, Universidade Estadual de Santa Cruz, Ilhéus, BA, Brazil

[19]Estación de Biodiversidad Tiputini, Colegio de Ciencias Biológicas y Ambientales, Universidad San Francisco de Quito, Quito, Ecuador

[20]Wildlife Ecology and Conservation, University of Florida, Gainesville, FL, USA

[21]Laboratório de Biogeografia da Conservação e Macroecologia, Instituto de Ciências Biológicas, Universidade Federal do Pará, Belém, PA, Brazil

[22]Centro Nacional de Pesquisa e Conservação de Mamíferos Carnívoros (CENAP/ICMBio), Atibaia, SP, Brazil

[23]Departamento de Ciências Biológicas, Escola Superior de Agricultura 'Luiz de Queiroz', Universidade de São Paulo, Piracicaba, SP, Brazil

[24]Instituto Pró-Carnívoros, Atibaia, SP, Brazil

[25]Panthera, Cali, Colombia

[26]Programa de Pós-Graduação em Ecologia, Instituto de Ciência Biológicas, Universidade Federal do Pará, Belém, PA, Brazil

[27]Laboratório de Ecologia de Mamíferos, Centro de Ciências Biológicas e da Natureza, Universidade Federal do Acre, Rio Branco, AC, Brazil

[28]Centro de Estudos da Biodiversidade, Universidade Federal de Roraima, Boa Vista, RR, Brazil

[29]Department of Ecology and Natural Resource Management, Norwegian University of Life Sciences, Ås, Norway

[30]Nicholas School of Environment, Duke University, Durham, NC, USA

[31]Faculty of Environmental Sciences and Natural Resource Management, Norwegian University of Life Sciences, Norway

[32]Laboratório de Biodoversidade Animal, Universidade Federal de Jataí, Jataí, GO, Brazil

[33]Engenharia Florestal, Universidade Federal de Viçosa, Viçosa, MG, Brazil

[34]Instituto Federal de Educação, Ciência e Tecnologia do Acre, Rio Branco, AC, Brazil

[35]Instituto Chico Mendes de Conservação da Biodiversidade, Brasília, DF, Brazil

[36]Instituto de Pesquisas Ecológicas (IPÊ), Nazaré Paulista, SP, Brazil

[37]Department of Mastozoology – Museu Paraense Emílio Goeldi, Belém, PA, Brazil

[38]Laboratório de Ecologia e Zoologia de Vertebrados, Instituto de Ciências Biológicas, Universidade Federal do Pará, Belém, PA, Brazil

[39]Earth Sciences and Ecology Department, Museu Paraense Emilio Goeldi, Belém, PA, Brazil

[40]Serviço da Estação Científica Ferreira Penna, Coordenação de Pesquisa e Pós-Graduação, Museu Paraense Emílio Goeldi, Belém, PA, Brazil

[41]Programa de Pós-Graduação em Ecologia e Manejo de Recursos Naturais, Universidade Federal do Acre, Rio Branco, AC, Brazil

[42]Grupo de Pesquisa em Ecologia de Vertebrados Terrestres (ECOVERT), Instituto de Desenvolvimento Sustentável Mamirauá, Tefé, AM, Brazil

[43]Comunidad de Manejo de Fauna Silvestre en la Amazonía y en Latinoamérica (ComFauna), Iquitos, Peru

[44]S.P.E.C.I.E.S. – The Society for the Preservation of Endangered Carnivores and their International Ecological Study, Ventura, CA, USA

[45]Center for Tropical Research, Institute of the Environment & Sustainability, University of California – Los Angeles, CA, USA

[46]Wildlife Conservation Society, Global Conservation Program, Bronx, NY, USA

(iD) DGR, 0000-0002-0100-3102; DMG, 0000-0002-5196-1384; MGML, 0000-0002-2203-7598

The persistent high deforestation rate and fragmentation of the Amazon forests are the main threats to their biodiversity. To anticipate and mitigate these threats, it is important to understand and predict how species respond to the rapidly changing landscape. The short-eared dog *Atelocynus microtis* is the only Amazon-endemic canid and one of the most understudied wild dogs worldwide.

We investigated short-eared dog habitat associations on two spatial scales. First, we used the largest record database ever compiled for short-eared dogs in combination with species distribution models to map species habitat suitability, estimate its distribution range and predict shifts in species distribution in response to predicted deforestation across the entire Amazon (regional scale). Second, we used systematic camera trap surveys and occupancy models to investigate how forest cover and forest fragmentation affect the space use of this species in the Southern Brazilian Amazon (local scale). Species distribution models suggested that the short-eared dog potentially occurs over an extensive and continuous area, through most of the Amazon region south of the Amazon River. However, approximately 30% of the short-eared dog's current distribution is expected to be lost or suffer sharp declines in habitat suitability by 2027 (within three generations) due to forest loss. This proportion might reach 40% of the species distribution in unprotected areas and exceed 60% in some interfluves (i.e. portions of land separated by large rivers) of the Amazon basin. Our local-scale analysis indicated that the presence of forest positively affected short-eared dog space use, while the density of forest edges had a negative effect. Beyond shedding light on the ecology of the short-eared dog and refining its distribution range, our results stress that forest loss poses a serious threat to the conservation of the species in a short time frame. Hence, we propose a re-assessment of the short-eared dog's current IUCN Red List status (Near Threatened) based on findings presented here. Our study exemplifies how data can be integrated across sources and modelling procedures to improve our knowledge of relatively understudied species.

## 1. Introduction

Understanding the factors that govern species distribution is one of the main goals of ecology [1]. Considering the apparent relative uniformity of the Amazonian ecosystem and the high number of species that it holds, understanding the drivers of Amazonian species distribution is particularly intriguing. From a conservation perspective, it has never been so urgent to understand the drivers of species distributions, given the recent human-induced environmental changes in Amazonia. Annual deforestation rates are persistent, and currently increasing within the Amazon [2]. This scenario is predicted to worsen with the local governments' ongoing plans for large-scale agriculture, cattle ranching and infrastructure expansion in the Amazon countries [3–5]. Moreover, global warming is expected to cause intensification of forest loss, which will be detrimental to biodiversity of the Amazon [6–8]. Besides identifying the relevant drivers for species distribution, it is important to understand how drivers act across different scales to effectively explain and predict distribution patterns. Despite widespread recognition of the importance of multi-scale approaches for species distribution modelling, the majority of published studies does not address multiple spatial scales [9].

The only Amazonian endemic canid, the short-eared dog *Atelocynus microtis*, is one of the least studied wild canid species worldwide [10]. The species is medium-sized (9–10 kg), elusive and its distribution remains unclear and poorly documented. Historical records of short-eared dogs extend from southeast Colombia to the central lowland of Bolivia, and from east Ecuador to Pará state in eastern Brazil, where confirmed records are limited to the right margin of the Amazon River. As the occurrence of the species in vast areas of the Amazon remains uncertain, some have suggested that its distribution might be patchy [11]. The majority of currently available information on the short-eared dog is from opportunistic or anecdotal records [12–14]. The limited literature available suggests that the species is mostly diurnal [15,16], usually solitary [14] and probably occurs naturally at low densities [17]. The short-eared dog seems to be associated with lowland, continuous, undisturbed areas near water in the Amazon forest [18–20]. However, to date, no study has investigated the species' demography, habitat association or space use throughout its distribution (but see studies from Bolivia in [20–22]).

Globally categorized as 'Near Threatened' by the IUCN Red List of Threatened Species [23], the short-eared dog was classified as 'Vulnerable' in the most recent assessment of the species' status in Brazil [10] and Peru [24]. The loss of prey, diseases transmitted by domestic dogs and, particularly, continued habitat loss are thought to be the major threats to the species' persistence [23]. A previous study suggested that about 40% of the species' distribution range lies in the 'Arc of Deforestation' [10], the region with the highest annual deforestation rates in Brazil, along the southern and southeastern border of the Amazon forest.

In this study, we integrate data across multiple spatial scales to assess habitat associations and conservation status of the short-eared dog. To that end, we used the largest dataset ever compiled of short-eared dog records in combination with species distribution models and environmental variables (regional scale) to predict shifts in the species' distribution under future deforestation scenarios. Additionally, we used occupancy models to investigate how habitat use by the short-eared dog is affected by attributes related to forest cover at a fine spatial scale (local scale). The results enabled us to propose adjustments to the short-eared dog's mapped distribution, identify environmental variables important for its occurrence, assess the likely magnitude of the effect of future deforestation on the species and suggest re-evaluation of its IUCN conservation status.

Similar to the short-eared dog [10], about one-quarter of the Amazon's mammal species are predicted to lose considerable amounts of their habitats [25]. Like most Amazonian species, the short-eared dog remains understudied, but with the increasing popularity of camera trapping [26], reliable geo-referenced data of many rare and cryptic mammal species are becoming more readily available. When pooled across large spatial areas, such data can inform both large- and small-scale analyses of habitat associations. Finally, as a forest-dependent species, thought to be associated with undisturbed habitat, the short-eared dog shares threats with many other Amazonian mammals and conservation interventions directed at short-eared dogs may benefit a range of species [27]. Therefore, we used this species as an example of how data from multiple sources can be integrated across spatial scales to better describe species distribution and potentially inform conservation.

# 2. Methods

We investigated the short-eared dog's habitat associations on two spatial scales. First, we used species presence records from different sources and species distribution models to map species habitat suitability and estimate its potential distribution across the Amazon. Because species–habitat relationships can be scale-dependent [28], we further investigated on a fine spatial scale, how forest cover and fragmentation affect the species' habitat use, based on data from a systematic camera trap survey in the Southern Brazilian Amazon in combination with occupancy models [29].

## 2.1. Species distribution (regional scale)

### 2.1.1. Dataset compilation

We compiled a short-eared dog record database using three sources: (i) published studies that recorded the species, (ii) specimens held in scientific collections with information on their geographical origin, and (iii) unpublished camera trap studies that recorded the species. In addition, we included records based on sightings from researchers with notable experience on Amazon carnivores. The records database ($n = 307$) was converted into geographical coordinates with the WGS 84 map datum (electronic supplementary material, appendix S1).

For modelling current species distribution, we only used occurrences with GPS coordinates and recorded after the year 2000. We rarefied records by randomly filtering with a 10 km radius buffer to reduce spatial auto-correlation due to unbalanced spatial sampling effort [30]. Because there is no information available on the home range size of short-eared dogs, the filtering buffer was based on the radius of a circular area of the home range size (1042 ha) of the crab-eating fox *Cerdocyon thous*, a similar-sized canid, in the Amazon region [31]. The final dataset, used for distribution modelling, had 97 presence records.

### 2.1.2. Environmental variables

To model the short-eared dog current distribution, we selected 21 possible explanatory variables: 19 bioclimatic variables from the Worldclim database [32], elevation from SRTM imagery [33] and land cover from the ESA GlobCover Project 2009 (for the complete list of variables, as well as their sources and spatial resolution, see electronic supplementary material, table S1). We used all variables in 30 arc-seconds (approx. 1 km$^2$) of spatial resolution (resampled to a coarser resolution when needed) and used Pearson's correlation test to identify collinearity between pairs of variables. For any combination of covariates with a correlation coefficient $r > |0.7|$, we retained the one considered to have greater ecological relevance to the species [34]. The final variables selected for modelling included: Annual Mean Temperature (Worldclim code BIO01), Mean Diurnal Temperature Range

(BIO02), Isothermality (BIO03, which is equal to Mean Diurnal Temperature Range/Temperature Annual Range), Annual Precipitation (BIO12), Precipitation Seasonality (BIO15, which is the coefficient of variation of Annual Precipitation), Precipitation of Warmest Quarter (BIO18) and land cover. Although climate can directly affect species distributions, it may also act indirectly through its influence on the vegetation [35]. Climate variables are the average for the years 1970–2000, with temperature data in degree Celsius and precipitation data in millimetres. To our knowledge, more recent climate variables with comparable resolution were not available for the Amazon region.

For land cover, we used the GlobCover layer, which represents land cover in the year 2009 in 22 classes defined by the United Nations Land Cover Classification System (LCCS) [36]. We acknowledge that records post-2009 may have been from areas that our land cover dataset characterizes as forested that were, in fact, deforested after 2009. Most of the post-2009 records (65%) were from protected areas that are unlikely to have experienced deforestation. For Brazil, where more recent forest cover maps are available, only two post-2009 records fell into a recently deforested area. We are, therefore, confident that the 2009 land cover information adequately reflects forest cover for the considered records. Prior to modelling, we reclassified the GlobCover layer classes into forest, non-forest and deforested (for reclassification details, see electronic supplementary material, table S2), making it comparable to the simulated future Amazon basin forest cover [25], which we used to predict future short-eared dog distribution (see below). Soares-Filho et al. [25] predicted yearly deforestation patterns across the Amazon basin from 2002 to 2050 according to two alternative conservation scenarios, 'Business as Usual' and 'Governance'. Both scenarios considered the ongoing deforestation trends, but the Business as Usual scenario added the effect of paving a set of major roads. On the other hand, the Governance scenario imposed a 50% limit for deforested land within each basin's sub-region and assumed that 100% of the forests within protected areas were preserved intact (for details on the scenario description, refer to [25]). When contrasting Amazon basin forest cover modelled by Soares-Filho et al. [25] against current maps of deforestation in Brazil [2], these models have performed well in predicting deforestation, at least for Brazil, the country that encompasses most of the short-eared dog's distribution (see electronic supplementary material, figure S1).

We used the selected climate variables and the reclassified GlobCover layer (hereafter, current forest cover) to model the species' current habitat suitability and estimate its potential distribution. Then, we replaced the current forest cover layer with simulated future Amazon basin forest cover modelled by Soares-Filho et al. [25] to predict future short-eared dog habitat suitability and distribution.

One of the criteria used by the IUCN Red List to assess threatened species categories is reduction in population size over the last 10 years or three generations (whichever is longer). This reduction can be estimated or inferred from the decline in area of occupancy, extent of occurrence and/or quality of habitat [37]. As a consequence of being one of the least studied canids in the world, there is a lack of information on the biology of the short-eared dog, particularly on its demography and reproduction. Leite Pitman & Williams [23] proposed a 4-year generation length for the short-eared dog in the IUCN Red List assessment. Two years later, Leite Pitman & Beisiegel [10] updated the generation length to 6 years. Other canid species of similar body size have assumed generation length of 3–4 years (see the IUCN Red List assessment for species of the South American Lycalopex genus). In a tamed male short-eared dog, Leite Pitman observed no display of sexual features (e.g. testicles descended and complex call) until 3 years of age [38], which is late for a dog/fox. Here, we decided to use a 6-year generation length for predicting species future distribution as this is the most current expert estimate [10]. From a conservation perspective, adopting a longer generation length is a conservative approach, because opting for a shorter generation length implies that individuals are successful in reproducing earlier in life. Based on this generation time, we predicted its distribution under both (Business as Usual and Governance) conservation scenarios for the years 2027 and 2045, representing three and six short-eared dog generations into the future from 2009 (year of our current forest cover layer). We projected the predictive models for the extent of the Amazon basin region, except for the Guiana and Jaú-Negro interfluves, for which there are no short-eared dog records (electronic supplementary material, figure S2), in spite of existing wildlife research efforts.

### 2.1.3. Modelling procedures

We used the maximum entropy algorithm implemented in Maxent 3.4 [39,40] to estimate short-eared dog habitat suitability. This algorithm is a robust and widely used modelling framework that seeks out non-random relationships between a set of explanatory environmental variables and species occurrence records by using presence-only data, which are compared to a sample of random background

locations where presence is unknown [30]. This relationship is parametrized using Bernoulli generalized linear models with a complementary log–log (cloglog) link function [41]. One important assumption for Maxent is that sampling is representative of environmental conditions in the modelled area. Even though our presence-only species data were not collected using systematic surveys, we believe we meet this assumption, given that covariate values at species record locations cover most of the distribution of covariates across our area of inference (see electronic supplementary material, figure S3).

We used 70% of the dataset for training and 30% for testing the model. We generated subsets by applying bootstrapping methods with 10 random partitions with replacements [42]. For all runs, we used Maxent default setting with 500 iterations and 10 000 background points [43]. We used the area under the receiver operating characteristic curve (AUC) to measure the predictive performance of the model, which ranges from 0.5 for models that are no better than random to 1.0 for models with the perfect predictive ability [42]. Model performance is considered acceptable when AUC ≥ 0.7 [44]. Also, we evaluated the omission rate (false negative predictions by the model) for the threshold selected (see details on threshold selection below), the statistical significance of the omission rate, the relative contributions (%) of the environmental variables to our models and the response curves of the most influential environmental variables (accounting for more than 50% of the relative importance of variables) for the Maxent prediction.

Species distribution models predict a continuous value of habitat suitability (ranging from 0, unsuitable, to 1, highly suitable) for every pixel within the area of modelling inference, taking into account a set of environmental variables. However, to estimate species distribution area, it is necessary to convert continuous habitat suitability values into a binary variable of unsuitable (0) or suitable (1) by using a threshold. There are many approaches to determining thresholds, and based on Liu *et al*. [45], we selected three approaches: minimum training presence logistic threshold = 0.1061; 10 percentile training presence logistic threshold = 0.3102; and maximum test sensitivity plus specificity logistic threshold = 0.3994 from the Maxent output of the average model (out of 10 runs). We generated current distribution models using these three thresholds and invited 10 Amazon carnivore experts to indicate which model best represented the short-eared dog's current distribution, without knowing the habitat suitability threshold values. The majority of the experts (60%) selected the model based on the minimum training presence logistic threshold. Hence, we adopted the expert validated threshold to estimate the current and future distribution of the short-eared dog.

We assessed the current and future total area predicted to be occupied by the short-eared dog per country (including all countries in which the species has been recorded), per interfluve (Amazon basin subregions limited by the main rivers, see electronic supplementary material, figure S4) and within/outside protected areas and indigenous land (hereafter collectively referred to as protected areas). We used country-level spatial data from the Global Administrative Area (www.gadm.org), spatial data for protected areas from the World Database of Protected Areas [46] and interfluve boundaries from Ribas *et al*. [47]. Considering only the areas currently thought to be occupied by the short-eared dog that remain occupied in the future, we assessed the total area expected to have a decline higher than 50% on the habitat suitability per country, per interfluve and within/outside protected areas.

## 2.2. Habitat use (local scale)

### 2.2.1. Study area

The local scale study area covers 70 000 km² in the Arc of Deforestation in the Southern Brazilian Amazon (60°48′ W, 7°22′ S; 60°25′ W, 10°51′ S), in the states of Amazonas, Rondônia and Mato Grosso. The landscape consists of a mosaic of land cover types, comprising pristine Amazon forest, patches of natural savannah and fragmented forested areas. The climate in the region is tropical humid, with average monthly temperatures between 24 and 28°C. Average annual precipitation varies from 2060 to 2890 mm, with a rainy season from October to March and a dry season from April to September [48].

### 2.2.2. Camera trap survey and environmental variables

We compiled data from four camera trap surveys, adding up to 180 sites surveyed within the study area between February 2016 and June 2017. We allocated the camera trap sites to represent a range of land cover including continuous forest, open savannahs, transitional forest-savannah areas and forested patches under the anthropogenic impact (figure 1). Each site had one unbaited camera trap (models

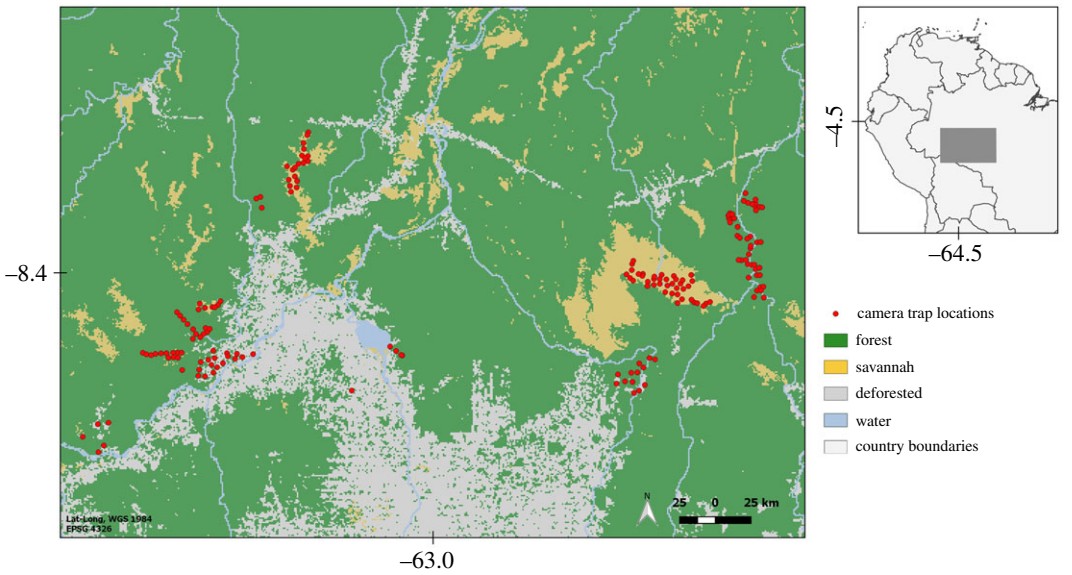

**Figure 1.** Map of the study area surveyed to investigate short-eared dog (*A. microtis*) habitat use in the Southern Brazilian Amazon, with camera trap locations and different vegetation covers.

PC800 Hyperfire, Reconyx or HD Essential/Aggressor, Bushnell; randomly assigned to sites) continuously active for at least 30 days (mean = 48, range = 32–92), totalling 8698 camera trap days. At each site, we set up the camera trap in locations with signs of medium- and large-sized mammals, at a height of approximately 30 cm above the ground. Camera traps operated 24 h d$^{-1}$, with no delay between subsequent potential triggers, and on average 3640 m (range = 815–40 547 m) between sites.

We used independent short-eared dog records (photos at the same site at least 24 h apart) to create a detection history (a site-by-occasion matrix; 0 for non-detection, 1 for detection). We investigated the effect of three site covariates on short-eared dog habitat use: mean of forest cover, forest edge density (total perimeter of forest patches divided by area) and forest patchy density (number of forest patches divided by area). The latter two variables are metrics of forest fragmentation. Because they were calculated based on a binary layer (forest/non-forest), they do not differentiate between deforested and naturally open habitats. We were unable to include habitat type (i.e. savannah versus forest) as a predictor variable, because short-eared dogs were very rarely recorded in open savannahs. Because species may respond to different landscape characteristics at different scales [49], we measured all site covariates at six different radii (0.5, 1, 1.5, 3, 5 and 10 km) around each camera trap. We used the Global Forest Change (GFC, [50]) layer for year 2016 to represent forest cover. In the GFC layer, pixel values are percentage forest cover (0–100%) in 1 arc-second (approx. 30 m) resolution, where forest is defined as canopy closure for all vegetation taller than 5 m. We calculated the fragmentation metrics edge density and patch density using the 'r.li' module in GRASS GIS 7.0 [51]. We used a binary forest/non-forest layer based on the reclassified 2016 GFC forest cover layer (pixels greater than or equal to 75% were considered forested; [52]) as the input layer to calculate fragmentation metrics (as performed by Tan *et al*. [52]). We opportunistically tested the effects of distance to water, distance to roads, elevation and human footprint index on the short-eared dog habitat use. However, we report these results in electronic supplementary material, table S8 because our main interest was understanding the effects of forest composition and configuration. We considered camera trap effort (i.e. number of days camera traps were active during each 20-day sampling occasion) as a survey covariate.

### 2.2.3. Modelling procedures

We modelled short-eared dog habitat use using occupancy models. Because short-eared dogs are mobile individuals and our sample units (camera trap locations) are probably smaller than the presumed home range size of the species, we adopted the terminology 'site use' or 'habitat use' instead of 'site occupancy'. Prior to the analysis, we collapsed the detection history into 20-day sampling occasions. Collapsing sampling periods increased temporal independence among occasions and overall detection probability, which can prevent model convergence if too low [53]. Covariates were usually correlated

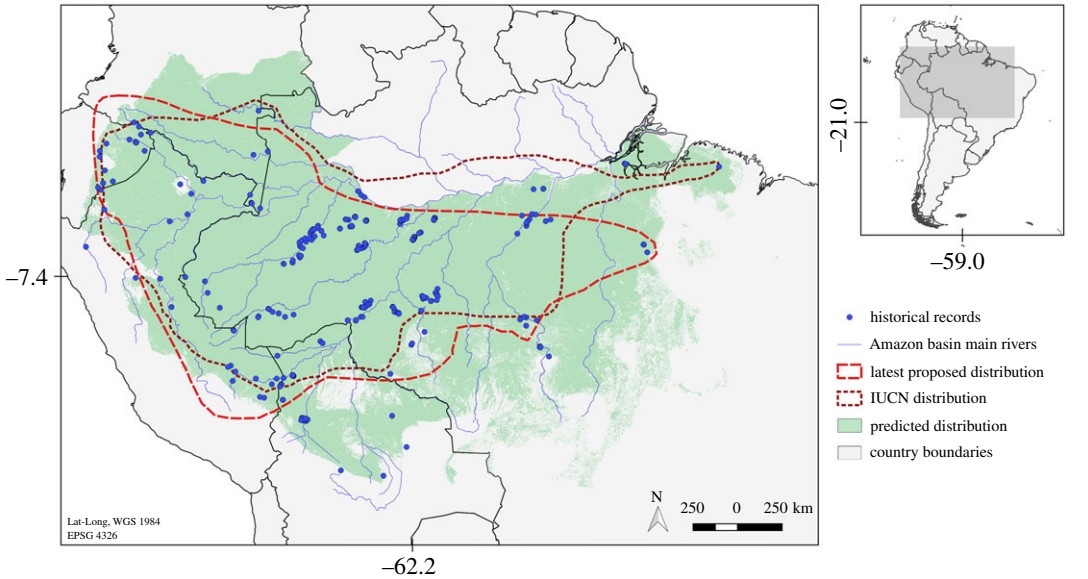

**Figure 2.** Short-eared dog (*A. microtis*) record locations compiled by this study and three alternative proposed species distributions: the IUCN distribution [23], the latest proposed distribution [10] and the current distribution predicted by this study based on Maxent model with land cover and climate covariates.

across different buffer sizes. Therefore, for each site covariate, we ran single-species, single-season, single-covariate occupancy models for each buffer size and retained the site covariate for the buffer size with the lowest AIC value (scale-optimization; [9]). If the null model performed better than any buffer size for a given variable, we did not include that variable in the next modelling steps. Then, we used covariates at their best performing buffer size to run multi-covariate occupancy models for all possible combinations of selected variables. We included the camera trap effort as a detection covariate for all competing models. We conducted all spatial data processing in Quantum GIS (QGIS) v. 2.14.19 [54] and ran occupancy models in the R package 'unmarked' v. 0.10–6 [55] in R v. 3.1.2 [56].

# 3. Results

## 3.1. Species distribution

We assembled 307 short-eared dog occurrence records (electronic supplementary material, appendix S1), of which 191 were previously unpublished records. Our dataset populates and expands previously proposed global distributions for the species, as some of the compiled records fall outside the distribution adopted by the IUCN [23] or proposed by Leite Pitman *et al*. [57] (figure 2).

The current species distribution model performed well (AUC = 0.785 ± 0.039 s.e., omission = 5%, $p \leq$ 0.001) and predicted the species to be distributed over an area of 4 302 532 km$^2$ (table 1), of which 1 882 654 km$^2$ (43.8%) were within protected areas. The most influential variables for the species' current distribution were land cover (28.78%), annual precipitation (BIO12, 19.84%) and precipitation seasonality (BIO15, 19.01%) (figure 3). Within the predicted current distribution area, the annual precipitation varied between 130 and 7096 mm (mean = 2590, s.d. = 547 mm), precipitation seasonality (its coefficient of variation) ranged from zero to 79 (mean = 44, s.d. = 16) and approximately 98% was covered by forest.

The predicted future short-eared dog distributions showed reductions in the area relative to the current distribution from 13.8% to 17.7% by 2027, and from 14.8% to 21.2% by 2045, considering the Governance and Business as Usual scenarios, respectively (figure 4). The predicted reduction in the short-eared dog distribution area outside protected areas was at least twice as high as the reduction within protected areas, regardless of the conservation scenario (electronic supplementary material, table S3).

Even though the confirmed records of the short-eared dog spanned five countries (Brazil, Bolivia, Colombia, Ecuador and Peru), approximately 80% of the predicted current distribution were in only two countries: Brazil (65%) and Peru (14.7%). The extent of predicted distribution area loss in Brazil was one or two orders of magnitude higher than in any other country (electronic supplementary

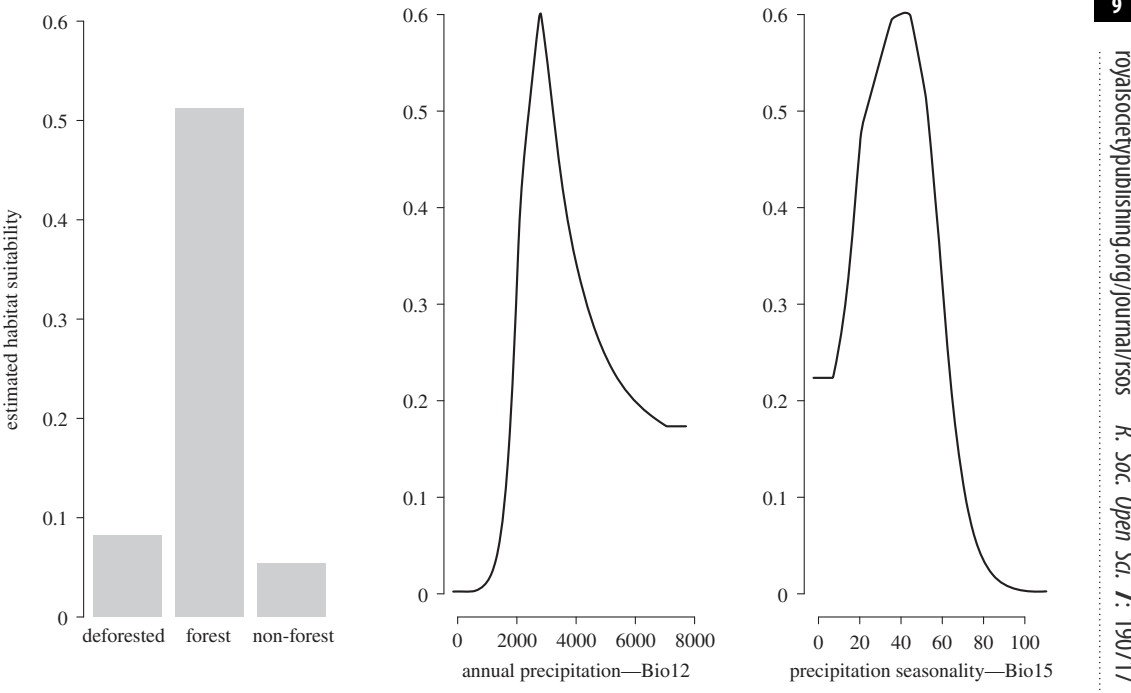

**Figure 3.** Relationship between habitat suitability for the short-eared dog (*A. microtis*) with the most important environmental variables, based on Maxent predictions of the species' current distribution (*n* = 97 presence records).

**Table 1.** Predicted area (km$^2$) of the short-eared dog (*A. microtis*) distribution by 2027 and 2045 taking into account two alternative conservation scenarios, Governance and Business as Usual, for the Amazon basin (following [25]). Distribution areas were estimated with Maxent using land cover and climatic variables. Areas were calculated within and outside protected areas (PA), as well as for each country with species records, and for different Amazon basin interfluves.

| | current distribution | Governance | | Business as Usual | |
|---|---|---|---|---|---|
| | | 2027 | 2045 | 2027 | 2045 |
| total | 4 302 532 | 3 708 668 | 3 669 660 | 3 543 121 | 3 389 235 |
| within PA | 1 882 654 | 1 742 794 | 1 737 404 | 1 704 370 | 1 638 808 |
| outside PA | 2 419 877 | 1 965 873 | 1 932 257 | 1 838 751 | 1 750 427 |
| *per country* | | | | | |
| Colombia | 425 293 | 407 310 | 403 682 | 404 527 | 399 103 |
| Ecuador | 85 015 | 75 097 | 69 976 | 73 283 | 64 709 |
| Peru | 633 960 | 583 941 | 575 330 | 582 791 | 569 401 |
| Bolivia | 361 711 | 272 661 | 267 113 | 269 469 | 256 793 |
| Brazil | 2 793 245 | 2 367 289 | 2 351 201 | 2 210 778 | 2 096 974 |
| *per interfluve* | | | | | |
| Pará | 77 769 | 51 321 | 51 278 | 48 091 | 47 981 |
| Xingu | 282 455 | 205 919 | 203 984 | 164 744 | 146 221 |
| Tapajós | 508 120 | 427 967 | 420 647 | 383 399 | 338 979 |
| Rondônia | 545 231 | 488 754 | 485 370 | 451 190 | 425 143 |
| Inambari | 1 349 937 | 1 282 824 | 1 277 950 | 1 282 146 | 1 273 091 |
| Imeri-Napo | 856 195 | 828 880 | 822 226 | 824 737 | 813 362 |

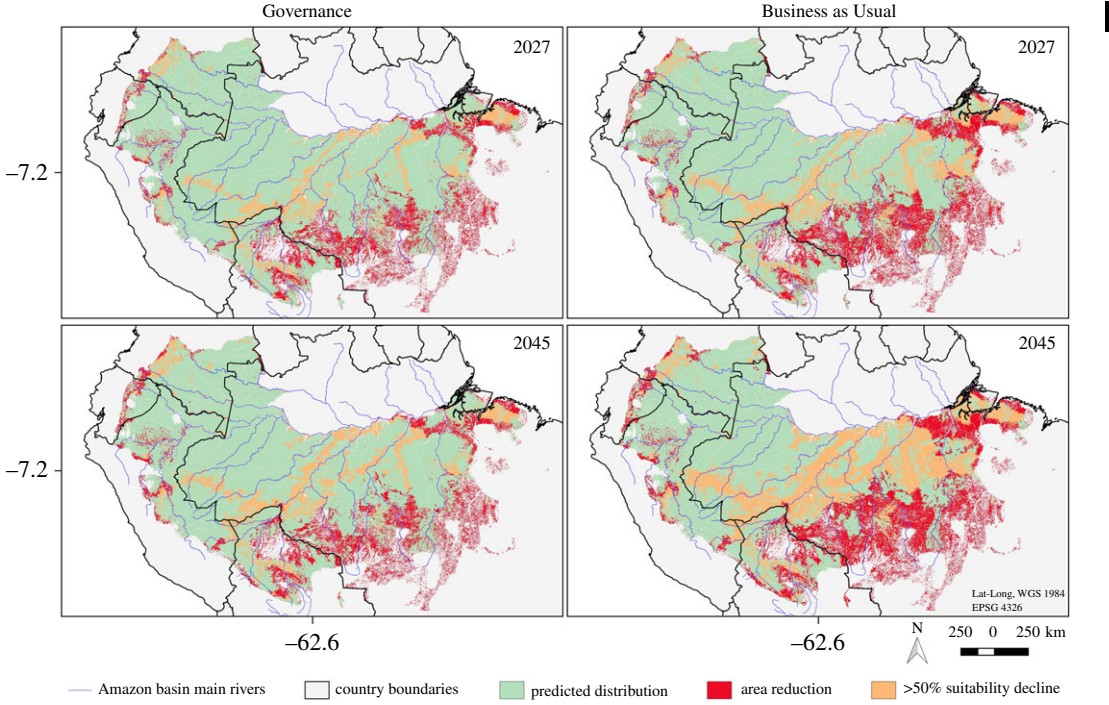

**Figure 4.** Short-eared dog (*A. microtis*) predicted distributions based on Maxent model with land cover and climate covariates. The maps depict the area within the species' current predicted distribution that is expected to be lost or have a reduction in habitat suitability greater than 50% by 2027 and 2045, taking into account two alternative deforestation scenarios ('Governance' and 'Business as Usual' for the Amazon basin (following [25]). Years 2027 and 2045 represent three and six short-eared dog generations time into the future from the current model (2009).

material, table S4). However, proportionally, Bolivia was expected to have the highest loss of short-eared dog distribution area in any considered conservation scenario (electronic supplementary material, table S3). There was a directional pattern of predicted distribution area loss across interfluves. The far eastern interfluves (Pará and Xingu) are predicted to experience the highest loss of distribution area. This trend declines from east to west, with the far western interfluves (Inambari and Imeri-Napo) having the smallest reduction in distribution area (electronic supplementary material, tables S3 and S4).

The proportions of the area expected to have a sharp decline (greater than 50% of the current value) in species habitat suitability followed a pattern similar to the loss of distribution per country, per interfluve and whether or not the area is protected (electronic supplementary material, tables S5 and S6). That leads to drastic consequences to the species when considering the combined effect of distribution area and habitat suitability reductions. For instance, for any given scenario, more than 30% of the short-eared dog's current distribution outside protected area is expected to be affected (distribution area lost or greater than 50% reduction in habitat suitability) by forest loss. An extreme case is the Pará interfluve, where at least 72% of the species' current distribution is expected to be lost or experience a sharp decline in habitat suitability by 2027 (table 2).

### 3.1.1. Habitat use

The camera trap survey in the Southern Brazilian Amazon produced 29 independent records of short-eared dogs at 19 different sites, resulting in a trapping success of 0.33 independent records per 100 trap days and naive occupancy of 0.1.

Based on the AIC values of the single-covariate models, the best performing buffer radius was 0.5 km for forest cover, and 1 km for edge density. For patch density, the null model (no patch density variable included as state variable predictor) performed best. Therefore, we did not include patch density in the next modelling steps (results of single-covariate models can be found in electronic supplementary material, table S8).

When combining the two selected site covariates, the best-ranked models (models with ΔAIC < 2) included forest cover and edge density, indicating that both variables are important to predict the species' habitat use (table 3). Therefore, we report the results for the model that included both covariates.

**Table 2.** Proportion of area (%) of short-eared dog (*A. microtis*) distribution expected to be lost or have a reduction of habitat suitability greater than 50% by 2027 and 2045 under two conservation scenarios, Governance and Business as Usual, for the Amazon basin (following [25]). Distributions were estimated with Maxent using land cover and climatic variables. Values were calculated within and outside protected areas (PA), as well as for each country with species records, and for different Amazon basin interfluves. Values above 30% are marked with '*', and above 50% are highlighted in bold. Estimated areas in km$^2$ (instead of proportions) are available in electronic supplementary material, table S7.

| | Governance | | Business as Usual | |
|---|---|---|---|---|
| | 2027 | 2045 | 2027 | 2045 |
| total | 22.4 | 26.0 | 29.4 | 42.3* |
| within PA | 10.5 | 12.6 | 16.0 | 30.2* |
| outside PA | 31.6* | 36.4* | 39.8* | **51.6*** |
| *per country* | | | | |
| Colombia | 12.3 | 16.2 | 13.1 | 19.0 |
| Ecuador | 13.7 | 20.2 | 16.0 | 27.5 |
| Peru | 11.5 | 14.0 | 12.2 | 16.4 |
| Bolivia | 34.4* | 37.7* | 36.0* | 42.8* |
| Brazil | 25.1 | 28.8 | 35.3* | **52.0*** |
| *per interfluve* | | | | |
| Pará | **72.6*** | **72.7*** | **79.5*** | **79.8*** |
| Xingu | 35.4 | 36.1* | **58.6*** | **68.7*** |
| Tapajós | 23.6 | 27.3 | 40.9* | **66.3*** |
| Rondônia | 22.6 | 29.3 | 32.9* | **57.4*** |
| Inambari | 15.5 | 19.9 | 18.1 | 30.5* |
| Imeri-Napo | 6.8 | 9.0 | 7.4 | 11.0 |

**Table 3.** Results of AIC-based model selection for short-eared dog occupancy models in the Southern Brazilian Amazon. ΔAIC is the relative difference in AIC values compared with the top ranked model. The site covariates tested were forest cover in a 0.5 km buffer (FOR0.5), and forest edge density in a 1 km buffer (ED1). Effort (EFF) was a survey covariate included in all models.

| model | parameters | AIC | ΔAIC |
|---|---|---|---|
| p(EFF) psi(FOR0.5) | 4 | 178.7 | 0.00 |
| p(EFF) psi(FOR0.5 + ED1) | 5 | 180.6 | 1.94 |
| p(EFF) psi(ED1) | 4 | 184.8 | 6.12 |
| p(EFF) psi(.) | 3 | 186.2 | 7.50 |

Short-eared dog site use was positively associated with forest cover and negatively associated with forest edge density (figure 5). Detection probability varied from 0.19 to 0.22 depending on effort, and the mean predicted probability of site use was 0.21 (range: 0.01–0.36; s.d.: 0.12). The parameter estimates for the selected model are available in electronic supplementary material, table S9.

## 4. Discussion

Combining the most comprehensive dataset on short-eared dog occurrence with species distribution and occupancy models, our results make clear that forest loss poses a serious threat to the conservation of this species, even on a relatively short time scale. Previous short-eared dog distribution limits have been developed based on the best available information and expert consensus. However, we believe that the distribution proposed here (figure 2) improves on previous versions [10,18] and supports previous distributions for the Bolivian portion of the range [20,21]. Our analyses are based on a much larger

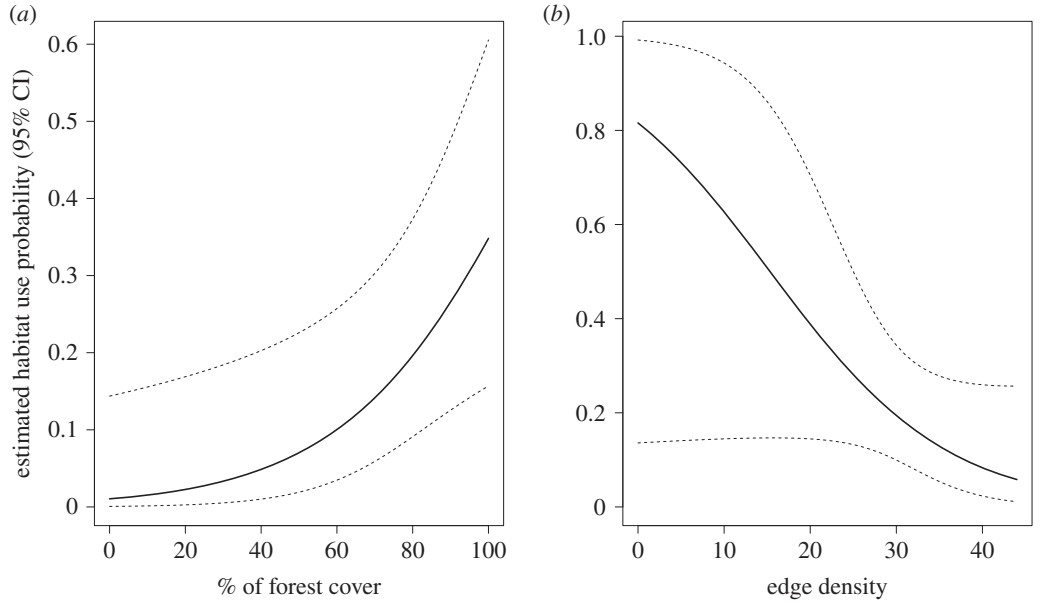

**Figure 5.** Relationship between estimated short-eared dog (*A. microtis*) site use probability and per cent of forest cover in a 0.5 km buffer (*a*) and forest edge density in a 1 km buffer (*b*) in the Southern Brazilian Amazon. Parameters were estimated with single-season occupancy models using camera trap data.

number of records, many of which were not available in the past, in combination with modelling procedures that account for biologically meaningful environmental variables and expert judgements. Moreover, natural features limit the borders of this newly proposed distribution: the Andes to the west, the Amazon River to the north and the absence of forest cover in the south and east. All recent short-eared dog records that fall outside the previously proposed species distribution were encompassed within the revised distribution that our model suggests.

According to our predicted current distribution model, the short-eared dog potentially occurs over an extensive (4 302 532 km$^2$) and continuous area, through most of the Amazon region south of the Amazon River. Despite reliable reports of the species swimming in rivers [23], and some recent records on the edge of the right margin of the Solimões River, the short-eared dog has not been confirmed on the left margin of the Amazon River. Even if individuals can occasionally cross the Amazon River, there is no evidence that the short-eared dog has established a population on the left margin, since long-term camera-trapping monitoring projects along the left margin have never detected the species (E.R. and A.A. 2018, personal communication).

Forest cover was the most influential environmental variable for short-eared dog habitat suitability in our model, confirming the species' reputed association with forested areas [19]. Unfortunately, high-resolution spatial layers that depict different forest structures are not available for the Amazon. However, variation in climate variables might indicate variation in forest structure. Two precipitation variables were important in our model. Short-eared dog habitat suitability is expected to be close to zero where annual precipitation is below 1500 mm and in areas with highly variable annual precipitation (which implies the presence of a severe dry season). Habitat suitability tended to peak at annual precipitation around 3000 mm, and at intermediate levels of precipitation variability (figure 3). Precipitation in Amazonia has declined during the past century [58], increasing the length and intensity of the dry season [59]. Some global climate models project significant further drying of the Amazon in the future. Our results suggest that this change in climate might add to the predicted range loss of the short-eared dog due to forest loss.

Our predictions indicate that forest loss could cause significant reductions in species distribution range and habitat suitability. The two alternative conservation scenarios differed considerably in the intensity of their effects. As expected, the Business as Usual scenario had stronger negative impacts than the Governance scenario, such that distribution losses predicted for 2045 under the Governance scenario are comparable with the distribution losses predicted in 2027 under the Business as Usual scenario.

Both scenarios modelled by Soares-Filho *et al.* [25] are conservative as they do not consider climate change effects that are expected to cause substitution of forests by savannah-like vegetation in the Amazon by the end of this century [60,61] or earlier [62]. For jaguars (*Panthera onca*), for example,

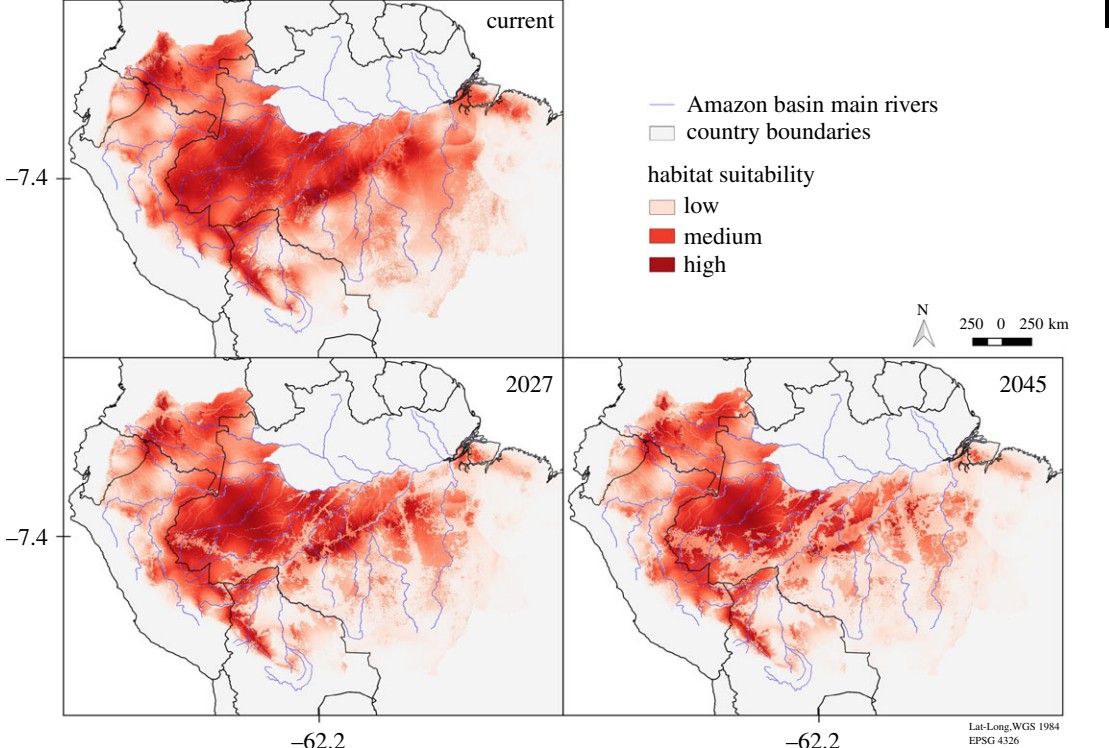

**Figure 6.** Short-eared dog (*A. microtis*) predicted habitat suitability based on Maxent model with land cover and climate covariates, according to the deforestation Business as usual scenario for the Amazon basin (following [25]) for year 2009 (current model), 2027 and 2045, which represents three and six short-eared dog generations time into the future from the current model.

Tôrres *et al*. [63] predicted a reduction in habitat suitability in the Amazon due to climate change. Also, these conservation scenarios do not consider forest degradation through logging and fire [25], which are severe threats to the Amazon forest [62]. Moreover, in the Governance scenario, deforestation rates are assumed to decline over time in response to external incentives and carbon market initiatives. However, after almost a decade of decline (2004–2012), annual deforestation rates in the Amazon started to climb again in 2013 [2], and so even the Business as Usual scenario might be too optimistic. In 2011, the Brazilian Federal Law of Forest Protection [64] was drastically relaxed, allowing reductions to the amount of forest that must be preserved on private lands, suspending the obligation of having any preserved areas for small-sized properties and offering amnesty from penalties for previous illegal cuts [65]. Such regulatory changes are already affecting deforestation rates [66] and were not accounted for in the models of Soares-Filho *et al*. [25]. Our model predictions are also on the conservative side, as the species' current distribution is based on a low habitat suitability threshold. Higher thresholds would predict even greater short-eared dog distribution losses (electronic supplementary material, figure S5). Therefore, hereafter we will discuss the results based on the less-conservative Business as Usual scenario predictions (figure 6).

The reductions in distribution area and habitat suitability were considerably lower within protected areas, even for the Business as Usual scenario, which allows deforestation within protected areas. This might be because protected areas in the Amazon have some success in decreasing deforestation within their boundaries when compared with unprotected areas [67]. In addition, protected area locations worldwide are often biased towards places that are less likely to face land conversion pressures even in the absence of protection [68].

Among the countries where the short-eared dog is present, Peru and Brazil had the highest proportions of forest loss between 2000 and 2016 ([50], see summary in electronic supplementary material, table S10). These two countries were estimated to contain 80% of the short-eared dog distribution. Consequently, more than 1.2 million $km^2$ of the species' distribution are expected to be lost or have a drastic reduction of habitat suitability by 2027. This means that, within three generations, approximately 30% of the short-eared dog's current distribution is expected to be affected by forest loss. This proportion might reach 40% of the species' distribution in non-protected areas and exceed 60% in some interfluves. By 2045, more than 40% of the current species distribution is

expected to be affected by deforestation, and all interfluves, except the Imeri-Napo, will have at least 30% of the species distribution area lost or with drastic habitat suitability reduction. These estimates are based on a 6-year generation length, and there is considerable uncertainty about that quantity due to lack of biological data. If the generation length used here is overestimated, our predictions of habitat loss and degradation within three generations may be somewhat pessimistic. Thus, if short-eared dog's generation length is shorter than assumed in this study, our future prediction horizons would also be shorter and a smaller proportion of the species' distribution would be expected to be impacted by deforestation. However, predicted deforestation within the current distribution of the short-eared dog based on the Business as Usual model by Soares-Filho *et al*. [25] for an 18-year time frame (i.e. using a 6-year generation length) compared to a 12-year time frame (i.e. using a 4-year generation length) are very similar, with 24% and 21%, respectively. This suggests that even under a shorter generation length, the species can be expected to lose comparable amounts of its range over the course of three generations.

Forest cover also had an important effect on species space use at the local scale. The results of the occupancy model indicated that in addition to the positive effect of the presence of forest, the short-eared dog was negatively affected by the density of forest edges. Our estimate of this negative effect may be conservative, as our analysis does not distinguish between natural edges (open habitats) and anthropogenic edges, which are expected to be more detrimental. This relationship is concerning as deforestation and logging combined create more than 50 000 km of new forest edges every year in the Amazon [69]. Our local-scale analysis also indicated that the species seems to be responding to forest cover on a very fine scale, as variables performed better when calculated for small buffer radii (0.5 and 1 km, rather than 5 or 10 km). Overall, the estimated probability of site use and probability of detection were low for the species at our study site, confirming that the species is elusive and probably occurs at low density [19]. None of the other covariates we explored opportunistically (electronic supplementary material, table S8) were important predictors of short-eared dog space use; however, our study was specifically designed to assess effects of forest configuration, and therefore we cannot make strong inference on other potential drivers of the species' space use.

The estimated negative effects of forest loss on the short-eared dog's habitat suitability, distribution and space use will probably lead to a reduction in the overall population size, and possible fragmentation and isolation of small (sub-)populations. This process is likely to be ongoing already, as it has been reported for other species in the Arc of Deforestation [70]. Additionally, as human-dominated areas expand throughout the short-eared dog's distribution, the risk of human-induced mortality due to diseases from domestic animals [71] and direct killing increases. The species has been recorded scavenging on human refuse and some of the short-eared dog records reported here were roadkill or hunted individuals. Finally, habitat changes may favour more generalist canid species like the crab-eating fox [72], and resulting interspecific competition could put additional pressure on the short-eared dog.

Considering the magnitude of the predicted negative effects of forest loss on the short-eared dog, we suggest re-assessment of the species' current IUCN status (Near Threatened) throughout its distribution, as an upgrade of its threat category seems warranted. Our study provides evidence that the ongoing forest loss across the Amazon will negatively affect more than 30% of the species' distribution. This suggestion reinforces the most recent Brazilian national assessment for the short-eared dog, which categorized the species as 'Vulnerable' [10]. Once again, we highlight that our suggestion to re-assess the short-eared dog's IUCN status is based on a 6-year generation length. Shorter generation lengths might result in distribution/habitat loss that do not satisfy IUCN criteria for upgrading the species threat category.

Often, particularly in diverse forested ecosystems, individual studies yield sparse data on rare or elusive species. This study exemplifies how data can be integrated across sources and modelling procedures to improve our knowledge on understudied species, by providing important information on ecological aspects for one of the most understudied canids in the world and highlighting main threats to its persistence. This much needed information should be used to re-assess the species' threat category at national and global scales, to underpin land use planning and management that will contribute towards short-eared dog conservation in the increasingly developed and human-dominated Amazonian landscapes. The situation of the short-eared dog is no exception: the distributions of other Amazonian carnivores, such as ocelot (*Leopardus pardalis*) and jaguar, are significantly affected by forest cover in Amazonia [73,74], and the same patterns have been observed in other tropical forests [52,75]. Many other mammals, particularly primates, are also notably susceptible to forest loss and degradation [76,77]. For those species, the future is bleak and they all

share the urgent need to reduce deforestation, which requires political will and commitment at local, national and international levels [8].

Data accessibility. The dataset used in this study is available in the electronic supporting material.

Authors' contributions. D.G.R., K.M.P.M.B.F., L.G.S., C.K.W.T., F.G.L., C.O., D.M. and R.S. were involved in writing the manuscript and conducting analysis. All other authors provided data and revised the manuscript.

Competing interests. The authors have declared that no competing interests exist.

Funding. D.G.R. received a scholarship from the CAPES/Doutorado Pleno no Exterior (grant no. 88881.128140/2016-01). K.M.P.M.B.F. received a productivity fellowship granted by the National Council of Technological and Scientific Development (grant no. 308632/2018-4). E.V. received support from CNPq (grant no. 308040/2017-1). E.M.v.M. received support from Fundação Grupo Boticário de Proteção a Natureza and Fundação de Amparo à Pesquisa do Estado do Amazonas. L.G.S. received support from CNPq and FACEPE. M.A. received support from School of Environmental Science at University of East Anglia (UEA), the Explorers Club, Idea Wild, and The Alongside Wildlife Foundation. T.H. received support from Research Council of Norway and the Norwegian University of Life Sciences. R.L.P. received support from Wildlife Materials, Idealwild, Conservation Food and Health, Disney Conservation Fund, Frankfurt Zoological Society, Amazon Conservation Association, Conservation International, World Wildlife Fund, and National Geographic. C.A.P. received support from Darwin Initiative for the Survival of Species grant. A.P.A. received support from the Brazilian Council of Technological and Scientific Development (CNPq). D.M.B. was supported by National Geographic Society, Universidad San Francisco de Quito, Tiputini Biodiversity Station, University of Missouri–St. Louis, University of Florida, and Walton Expeditions.

Acknowledgements. We are grateful to many field assistants that helped with data collection. This study would not be possible without their hard work. We also thank S. Alves and T.J.D. Ranzi for collaborating with data and R.M. Rabelo and F.P. Silva for advice on the analysis. We are grateful for the support of many institutions, including Mamirauá Institute of Sustainable Development, Wildlife Conservation Society, Campos Amazonicos NP, Mapinguari NP, Consórcio Hidrelétrico Teles Pires and Biota—Projetos e Consultoria Ambiental, Pousada Cumaru, the Bolivian Biodiversity and Protected Area Directorate, Secretaria de Estado de Meio Ambiente do Acre, National Institute of Natural Resouces – Peru (INRENA), National Forest Service – Peru (SERFOR), Áreas Protegidas da Amazônia (ARPA) and Instituto de Pesquisas Ecológicas (IPÊ).

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
