## [Reviewer comments · Royal Society Open Science]

Review History

RSOS-190717.R0 (Original submission)

Review form: Reviewer 1

Is the manuscript scientifically sound in its present form?

Yes

Are the interpretations and conclusions justified by the results?

Yes

Is the language acceptable?

Yes

Is it clear how to access all supporting data?

Yes

Do you have any ethical concerns with this paper?

No

Have you any concerns about statistical analyses in this paper?

I do not feel qualified to assess the statistics

Recommendation?

Accept with minor revision (please list in comments)

Comments to the Author(s)

This is a very good piece of work, contributing additional knowledge on the current distribution of short-eared dogs and assessing future impact of current and projected deforestation rates. The data and methodology are well presented, the results are clear, and the discussion advances the state of knowledge of the distribution of this little known Amazonian canid.

I found your choice of future horizons as 2027 and 2045 poorly justified by a 6 year generation time which is not supported by any demographic data and clearly too high for a mid-size canid. Also, unclear why to go for a landcover database that is 10 years old, when more recent data is available?

Below I list a few comments that I hope will help improve the final version of this work.

L106. 'Interfluve' technical term that is not widely used outside technical literature. You should define on first mention

L110. You can make a recommendation for a Red List reassessment, but should not make the recommendation to what category, since that will only determined by challenging the RL criteria with your new data. And this paper does not formally do that. You may want to want to say 'upgrading... its Red List status...'

L130. It's odd to use work that is 20 and 10 years old to support expectations of climate change impact on Amazonian biodiversity. Some of those authors predictions may be already supported by data, of refuted. There must be more recent work you could consider.

L126. and References henceforth. Your reference manager seems to have mangled many authors names, sometimes including given names, or initials, or listing way too many authors. (eg L146 DEL Maria Renata Pereira. Surname is Leite Pitman; L151. DEL M.R.P.). Please be careful to review your citations carefully before resubmission.

L140. You are using a field guide as a reference for patchy distribution. There are more detailed references (eg IUCN Action Plan; Macdonald & Sillero canid book, Wallace work) to support this statement.

L170 scale-dependent, use hyphen

L194. Fine to use crab-eating foxes as proxy for HR, but you should look more broadly and cite other studies, not just Courtenay's work. How about hoary foxes?

L214 & 216. There should be at least an overlap between datasets. You are using 1970-2000 for climate, but 2009 for land cover. Why not use more recent climate data, since it is readily available? And why 2006 land cover and not a more recent dataset?

L232. Soares-Filho 2006 predictions. How good were they 13 years later? That will give you an inkling whether his predictions will be any good going forward.

L237. Red List criteria. You need to use v3.1 2nd edition (IUCN 2012, NOT IUCN 2001)

L238. Generation time cannot be 6 years, when most canids of similar or larger since have generation times of 2-3 years (The RL assessment used 4 years). The later will take you to the 10 year window to assess decline (or 12 years, if you use 4 years, but not sure that can be sustained, since there is not actual demographic data to support it). This will work for your 2027 scenario, if you start from the end of your database assemblage. Less clear why you also provide a 2045 scenario. I'd use 30 or 50 years for the long-term scenario, as a matter of personal preference.

Justify why use 2045.

For a RL reassessment you can look at past changes, or use inferences of future changes.

L254-257. Explanation a bit woolly. Tighten up.

L286. Finally, I found a definition of interfluval. Should be presented on first mention.

L287. Definition unclear. Are you saying that protected areas INCLUDE indigenous land? This will require different wording.

L304. This methodology needs greater detail. It does not tell me how many of those cameras were deployed at a given time. 180 cameras at the same time?, 60 moved to three different arrays at different times? Also how far apart were the cameras set up from each other?

L327-330. The binary forest layer, where does it come from? Or you produced it yourselves?

L347. Have you defined camera trap effort somewhere?

L368. This has to be predicted for 10 years (or 12, if you can justify a 4 year generation time). You should not use 6x3, since it is not biologically sound.

L378 change 'any' to 'either'

L416. This excluded the most recent RL assessment Leite & Williams 2011

L423. Not sure I understand why 'Most' recent, and not all? That happened to the exceptions?

Explain.

L485. I have an issue with the choice of time window. When the RL reassessment takes place, it will be based on a 10 year window (with a realistic generation time of 2-3 years). Therefore, for this paper to be informative you should provide prediction for the right time interval.

L510. You need to properly reference your mention of disease as a potential risk.

L513. No need to repeat scientific name.

L516-517. This is a worthwhile recommendation but it should be a recommendation to reassess the species status, in light of the new data and analysis. You can't pre-adjudicate this assessment will necessarily be to 'Vulnerable'. That will be the outcome of testing against the criteria. It is, after all, a quantitative process, not a qualitative one. Reword accordingly.

L574. Address lots of typos and formatting.

Review form: Reviewer 2

Is the manuscript scientifically sound in its present form?

Yes

Are the interpretations and conclusions justified by the results?

No

Is the language acceptable?

Yes

Is it clear how to access all supporting data?

Yes

Do you have any ethical concerns with this paper?

No

Have you any concerns about statistical analyses in this paper?

Yes

Recommendation?

Major revision is needed (please make suggestions in comments)

Comments to the Author(s)

Wild dogs at stake: Deforestation threatens the only Amazon endemic canid, the short-eared dog (*Atelocynus microtis*); Rocha et al.; RSOS

Rocha et al. present a fascinating study of the severely under-studied short-eared dogs, combining historic presence records, camera trapping and species distribution models to predict current and future distribution of the species. The authors also present a multi-scale perspective (regional and local), which sheds some light on the scale-specific factors affecting the species' association with habitat. Overall, the paper is very well written and the authors have done a commendable job— considering the effort it would have taken to coordinate, initiate collaborations and compile information about a rare species from multiple sources and institutions. I do have some concerns related to the local scale analysis (explained below).

Introduction:

Specific comments:

L127: Please revise the sentence: “Besides the alarming deforestation rates, there is growing concern...”

Methods:

General comments: If I understood correctly, the authors use all presence records (post-rarifying) from 2000 onwards. Does this include data from 2009 to 2018/19? If so, there would be some mismatch in the forest cover (i.e. potential habitat) and current distribution right? The forest cover layer is from 2009, but presence records are from subsequent years as well. I get that this may be a limitation because of access to updated forest cover data. But this issue needs to be acknowledged in the Methods section.

Local scale analysis: Given that that authors list habitat loss, prey decline, disease from domestic dogs and other anthropogenic factors (road related mortality) as factors that negatively impact short-eared dogs, I wonder why none of these covariates, or variables that could serve as surrogates of these factors, were not considered for modeling habitat use. The authors also list lowland forests, areas close to water etc. as purported “good habitats” for the focal species. The large-scale predicted distribution also clearly indicates importance of protected areas for the species to persist in the long term. Elevation, distance to protected area boundary, distance to water bodies, proximity to human settlement, human footprint index, encounters of hunters/logging machinery, encounters of domestic dogs etc. are all really good potential covariates that would make the local scale analysis very interesting while also better informing management/conservation. The three covariates used to model occupancy PSI may be too broad to have show any influence at the local scale; this is evident in the estimates of the slope coefficients. Both forest cover and edge density have high SE values (95% CI overlaps 0, so no statistical significance), suggesting weak effects of the predictors. I strongly suggest that the authors re-analyse the local-scale data incorporating some/all the covariates listed above.

Specific comments:

L335: revise the sentence to state that the sample units are “likely” smaller than the “presumed” home-range size of the species.

Results:

General comments: Move Table S4 to main manuscript. This is the primary result of your study. Similarly, combine three panels of Figures S5, S6, S7 and present it as one figure in main manuscript. The results related to local-scale analysis would change when the data are re-analysed. Please present revised results accordingly.

Discussion:

General comments: The section is well written for the most part, and makes a compelling case for revising the IUCN status of short eared dog. I think the paper would benefit from expanding the narrative, both, in the Introduction and in the Discussion sections. As it stands, the focus is a bit narrow. If at all the species does qualify to be considered a "flagship" for conservation of forest systems in the Amazon, then the entire Discussion should make a more convincing case by establishing links between the short-eared dogs, their prey, competing predators and humans. The authors go some length in doing so in the last paragraph. But the argument is lacking in that it only makes a passing mention of the benefit to other species.

Decision letter (RSOS-190717.R0)

17-Jul-2019

Dear Mr Rocha,

The editors assigned to your paper ("Wild dogs at stake: Deforestation threatens the only Amazon endemic canid, the short-eared dog (*Atelocynus microtis*)") have now received comments from reviewers. We would like you to revise your paper in accordance with the referee and Associate Editor suggestions which can be found below (not including confidential reports to the Editor). Please note this decision does not guarantee eventual acceptance.

Please submit a copy of your revised paper before 09-Aug-2019. Please note that the revision deadline will expire at 00.00am on this date. If we do not hear from you within this time then it will be assumed that the paper has been withdrawn. In exceptional circumstances, extensions may be possible if agreed with the Editorial Office in advance. We do not allow multiple rounds of revision so we urge you to make every effort to fully address all of the comments at this stage. If deemed necessary by the Editors, your manuscript will be sent back to one or more of the original reviewers for assessment. If the original reviewers are not available, we may invite new reviewers.

- Data accessibility

If you wish to submit your supporting data or code to Dryad (<http://datadryad.org/>), or modify your current submission to dryad, please use the following link:
<http://datadryad.org/submit?journalID=RSOS&manu=RSOS-190717>

- Competing interests

- Authors' contributions

- Acknowledgements

- Funding statement

Kind regards,

Andrew Dunn

on behalf of Professor Michael Bruford (Associate Editor) and Kevin Padian (Subject Editor)
openscience@royalsociety.org

Editor comments:

Thank you for your submission. As you will see the reviewers are generally happy with it but have a series of fairly minor concerns that you should be able to address handily. Best wishes in your revisions.

Comments to Author:

Reviewers' Comments to Author:

Reviewer: 1

Comments to the Author(s)

This is a very good piece of work, contributing additional knowledge on the current distribution of short-eared dogs and assessing future impact of current and projected deforestation rates. The data and methodology are well presented, the results are clear, and the discussion advances the state of knowledge of the distribution of this little known Amazonian canid.

I found your choice of future horizons as 2027 and 2045 poorly justified by a 6 year generation time which is not supported by any demographic data and clearly too high for a mid-size canid. Also, unclear why to go for a landcover database that is 10 years old, when more recent data is available?

Below I list a few comments that I hope will help improve the final version of this work.

L106. 'Interfluve' technical term that is not widely used outside technical literature. You should define on first mention

L110. You can make a recommendation for a Red List reassessment, but should not make the recommendation to what category, since that will only determined by challenging the RL criteria with your new data. And this paper does not formally do that. You may want to want to say 'upgrading... its Red List status...'

L130. It's odd to use work that is 20 and 10 years old to support expectations of climate change impact on Amazonian biodiversity. Some of those authors predictions may be already supported by data, of refuted. There must be more recent work you could consider.

L126. and References henceforth. Your reference manager seems to have mangled many authors names, sometimes including given names, or initials, or listing way too many authors. (eg L146 DEL Maria Renata Pereira. Surname is Leite Pitman; L151. DEL M.R.P.). Please be careful to review your citations carefully before resubmission.

L140. You are using a field guide as a reference for patchy distribution. There are more detailed references (eg IUCN Action Plan; Macdonald & Sillero canid book, Wallace work) to support this statement.

L170 scale-dependent, use hyphen

L194. Fine to use crab-eating foxes as proxy for HR, but you should look more broadly and cite other studies, not just Courtenay's work. How about hoary foxes?

L214 & 216. There should be at least an overlap between datasets. You are using 1970-2000 for climate, but 2009 for land cover. Why not use more recent climate data, since it is readily available? And why 2006 land cover and not a more recent dataset?

L232. Soares-Filho 2006 predictions. How good were they 13 years later? That will give you an inkling whether his predictions will be any good going forward.

L237. Red List criteria. You need to use v3.1 2nd edition (IUCN 2012, NOT IUCN 2001)

L238. Generation time cannot be 6 years, when most canids of similar or larger have generation times of 2-3 years (The RL assessment used 4 years). The later will take you to the 10 year window to assess decline (or 12 years, if you use 4 years, but not sure that can be sustained, since there is not actual demographic data to support it). This will work for your 2027 scenario, if you start from the end of your database assemblage. Less clear why you also provide a 2045 scenario. I'd use 30 or 50 years for the long-term scenario, as a matter of personal preference. Justify why use 2045.

For a RL reassessment you can look at past changes, or use inferences of future changes.

- L254-257. Explanation a bit woolly. Tighten up.
- L286. Finally, I found a definition of interfluvial. Should be presented on first mention.
- L287. Definition unclear. Are you saying that protected areas INCLUDE indigenous land? This will require different wording.
- L304. This methodology needs greater detail. It does not tell me how many of those cameras were deployed at a given time. 180 cameras at the same time?, 60 moved to three different arrays at different times? Also how far apart were the cameras set up from each other?
- L327-330. The binary forest layer, where does it come from? Or you produced it yourselves?
- L347. Have you defined camera trap effort somewhere?
- L368. This has to be predicted for 10 years (or 12, if you can justify a 4 year generation time). You should not use 6x3, since it is not biologically sound.
- L378 change 'any' to 'either'
- L416. This excluded the most recent RL assessment Leite & William 2011
- L423. Not sure I understand why 'Most' recent, and not all? That happened to the exceptions? Explain.
- L485. I have an issue with the choice of time window. When the RL reassessment takes place, it will be based on a 10 year window (with a realistic generation time of 2-3 years). Therefore, for this paper to be informative you should provide prediction for the right time interval.
- L510. You need to properly reference your mention of disease as a potential risk.
- L513. No need to repeat scientific name.
- L516-517. This is a worthwhile recommendation but it should be a recommendation to reassess the species status, in light of the new data and analysis. You can't pre-adjudicate this assessment will necessarily be to 'Vulnerable'. That will be the outcome of testing against the criteria. It is, after all, a quantitative process, not a qualitative one. Reword accordingly.
- L574. Address lots of typos and formatting.

Reviewer: 2

Comments to the Author(s)

Wild dogs at stake: Deforestation threatens the only Amazon endemic canid, the short-eared dog (*Atelocynus microtis*); Rocha et al.; RSOS

Rocha et al. present a fascinating study of the severely under-studied short-eared dogs, combining historic presence records, camera trapping and species distribution models to predict current and future distribution of the species. The authors also present a multi-scale perspective (regional and local), which sheds some light on the scale-specific factors affecting the species' association with habitat. Overall, the paper is very well written and the authors have done a commendable job— considering the effort it would have taken to coordinate, initiate collaborations and compile information about a rare species from multiple sources and institutions. I do have some concerns related to the local scale analysis (explained below).

Introduction:

Specific comments:

L127: Please revise the sentence: "Besides the alarming deforestation rates, there is growing concern..."

Methods:

General comments: If I understood correctly, the authors use all presence records (post-rarifying) from 2000 onwards. Does this include data from 2009 to 2018/19? If so, there would be some mismatch in the forest cover (i.e. potential habitat) and current distribution right? The forest cover layer is from 2009, but presence records are from subsequent years as well. I get that this may be a limitation because of access to updated forest cover data. But this issue needs to be acknowledged in the Methods section.

Local scale analysis: Given that that authors list habitat loss, prey decline, disease from domestic

dogs and other anthropogenic factors (road related mortality) as factors that negatively impact short-eared dogs, I wonder why none of these covariates, or variables that could serve as surrogates of these factors, were not considered for modeling habitat use. The authors also list lowland forests, areas close to water etc. as purported “good habitats” for the focal species. The large-scale predicted distribution also clearly indicates importance of protected areas for the species to persist in the long term. Elevation, distance to protected area boundary, distance to water bodies, proximity to human settlement, human footprint index, encounters of hunters/logging machinery, encounters of domestic dogs etc. are all really good potential covariates that would make the local scale analysis very interesting while also better informing management/conservation. The three covariates used to model occupancy PSI may be too broad to have show any influence at the local scale; this is evident in the estimates of the slope coefficients. Both forest cover and edge density have high SE values (95% CI overlaps 0, so no statistical significance), suggesting weak effects of the predictors. I strongly suggest that the authors re-analyse the local-scale data incorporating some/all the covariates listed above.

Specific comments:

L335: revise the sentence to state that the sample units are “likely” smaller than the “presumed” home-range size of the species.

Results:

General comments: Move Table S4 to main manuscript. This is the primary result of your study. Similarly, combine three panels of Figures S5, S6, S7 and present it as one figure in main manuscript. The results related to local-scale analysis would change when the data are re-analysed. Please present revised results accordingly.

Discussion:

General comments: The section is well written for the most part, and makes a compelling case for revising the IUCN status of short eared dog. I think the paper would benefit from expanding the narrative, both, in the Introduction and in the Discussion sections. As it stands, the focus is a bit narrow. If at all the species does qualify to be considered a “flagship” for conservation of forest systems in the Amazon, then the entire Discussion should make a more convincing case by establishing links between the short-eared dogs, their prey, competing predators and humans. The authors go some length in doing so in the last paragraph. But the argument is lacking in that it only makes a passing mention of the benefit to other species.

Author's Response to Decision Letter for (RSOS-190717.R0)

See Appendix A.

RSOS-190717.R1 (Revision)

Review form: Reviewer 1

Is the manuscript scientifically sound in its present form?

Yes

Are the interpretations and conclusions justified by the results?

No

Is the language acceptable?

Yes

Do you have any ethical concerns with this paper?

No

Have you any concerns about statistical analyses in this paper?

No

Recommendation?

Accept with minor revision (please list in comments)

Comments to the Author(s)

As stated in my original review this is a very good piece of work, contributing additional knowledge on the current distribution of short-eared dogs and assessing future impact of current and projected deforestation rates. The data and methodology are well presented, the results are clear, and the discussion advances the state of knowledge of the distribution of this little known Amazonian canid.

That assessment still stands. However, I was disappointed by the authors' reluctance to review the value of their guessed generation time for the species (not supported by any demographic data). A 6 year generation time does not reflect what we know about the reproductive biology of similar sized canids. Therefore a proxy generation time should be used, revised down to 3 years (max 3.5 years) to reflect that observed across similar size canids. Future horizons should change accordingly. The choice of such a conservative generational time weakens the authors' argument for uplisting the species Red List status.

Review form: Reviewer 2

Is the manuscript scientifically sound in its present form?

No

Are the interpretations and conclusions justified by the results?

No

Is the language acceptable?

Yes

Do you have any ethical concerns with this paper?

No

Have you any concerns about statistical analyses in this paper?

Yes

Recommendation?

Major revision is needed (please make suggestions in comments)

Comments to the Author(s)

Provided as a separate document (Appendix B).

Decision letter (RSOS-190717.R1)

14-Feb-2020

Dear Mr Rocha,

Manuscript ID RSOS-190717.R1 entitled "Wild dogs at stake: Deforestation threatens the only Amazon endemic canid, the short-eared dog (*Atelocynus microtis*)" which you submitted to Royal Society Open Science, has been reviewed. The comments of the reviewers are included at the bottom of this letter.

Please submit a copy of your revised paper before 08-Mar-2020. Please note that the revision deadline will expire at 00.00am on this date. If we do not hear from you within this time then it will be assumed that the paper has been withdrawn. In exceptional circumstances, extensions may be possible if agreed with the Editorial Office in advance. We do not allow multiple rounds of revision so we urge you to make every effort to fully address all of the comments at this stage. If deemed necessary by the Editors, your manuscript will be sent back to one or more of the original reviewers for assessment. If the original reviewers are not available we may invite new reviewers.

- Ethics statement

- Data accessibility

- Competing interests

- Authors' contributions

- Acknowledgements

- Funding statement

As described in our instructions to authors (<https://royalsociety.org/journals/authors/author-guidelines/>), we need the original source files of any figures and tables included with your revised manuscript, as well as an editable version of your paper for our production processes. In your revision, please ensure that you upload:

- An editable, clean version of your revised manuscript.
- A tracked changes version of your revised manuscript - highlighting the edits you have made.
- Each figure uploaded separately; PDF or EPS format preferred.

Finally, we note that the following two email addresses are invalid, so please ensure that these are updated in the ScholarOne system when submitting your revised manuscript:

a.calouro@bol.com.br
fabiano_melo@ufg.br

Kind regards,
Lianne Parkhouse
Editorial Coordinator
Royal Society Open Science
openscience@royalsociety.org

on behalf of Professor Michael Bruford (Associate Editor) and Kevin Padian (Subject Editor)
openscience@royalsociety.org

Associate Editor Comments to Author (Professor Michael Bruford):

The two referees who reviewed your manuscript have come back with additional comments, one

of which potentially requires additional analysis. Please consider these requests and carry out the additional analysis if necessary

Subject Editor Comments to Author:

One reviewer feels strongly that you did not address what appears to be an unrealistic estimate of generation time, given what is known of canids. These and other concerns will need to be addressed fully in your final version of submission; please be explicit and best wishes for your revisions.

Reviewer comments to Author:

Reviewer: 1

Comments to the Author(s)

As stated in my original review this is a very good piece of work, contributing additional knowledge on the current distribution of short-eared dogs and assessing future impact of current and projected deforestation rates. The data and methodology are well presented, the results are clear, and the discussion advances the state of knowledge of the distribution of this little known Amazonian canid.

That assessment still stands. However, I was disappointed by the authors' reluctance to review the value of their guessed generation time for the species (not supported by any demographic data). A 6 year generation time does not reflect what we know about the reproductive biology of similar sized canids. Therefore a proxy generation time should be used, revised down to 3 years (max 3.5 years) to reflect that observed across similar size canids. Future horizons should change accordingly. The choice of such a conservative generational time weakens the authors' argument for uplisting the species Red List status.

Reviewer: 2

Comments to the Author(s)

Provided as a separate document.

Author's Response to Decision Letter for (RSOS-190717.R1)

See Appendix C.

Decision letter (RSOS-190717.R2)

10-Mar-2020

Dear Mr Rocha:

On behalf of the Editors, I am pleased to inform you that your Manuscript RSOS-190717.R2 entitled "Wild dogs at stake: Deforestation threatens the only Amazon endemic canid, the short-eared dog (*Atelocynus microtis*)" has been accepted for publication in Royal Society Open Science

subject to minor revision in accordance with the referee suggestions. Please find the referees' comments at the end of this email.

Please note that we require active email addresses from all authors associated with the manuscript before we can accept a final version. Presently, the following email addresses are not accepting messages from Royal Society Open Science - please supply the editorial office with alternative emails for these individuals:

-- a.calouro@bol.com.br
-- fabiano_melo@ufg.br

The reviewers and Subject Editor have recommended publication, but also suggest some minor revisions to your manuscript. Therefore, I invite you to respond to the comments and revise your manuscript.

- Ethics statement

- Data accessibility

If you wish to submit your supporting data or code to Dryad (<http://datadryad.org/>), or modify your current submission to dryad, please use the following link:
<http://datadryad.org/submit?journalID=RSOS&manu=RSOS-190717.R2>

- Competing interests

- Authors' contributions

- Acknowledgements

- Funding statement

Because the schedule for publication is very tight, it is a condition of publication that you submit the revised version of your manuscript before 19-Mar-2020. Please note that the revision deadline will expire at 00.00am on this date. If you do not think you will be able to meet this date please let me know immediately.

on behalf of Professor Michael Bruford (Associate Editor) and Kevin Padian (Subject Editor)
openscience@royalsociety.org

Associate Editor Comments to Author (Professor Michael Bruford):
Associate Editor
Comments to the Author:

Thanks for your revision which I think places the MS close to acceptance. I understand your stance on the generation time estimate and I agree with you that this is justified. I would just like to see you address it a bit more in the discussion perhaps with a caveat that using 6 years might change the scenario somewhat, for transparency.

Author's Response to Decision Letter for (RSOS-190717.R2)

See Appendix D.

Decision letter (RSOS-190717.R3)

23-Mar-2020

Dear Mr Rocha,

It is a pleasure to accept your manuscript entitled "Wild dogs at stake: Deforestation threatens the only Amazon endemic canid, the short-eared dog (*Atelocynus microtis*)" in its current form for publication in Royal Society Open Science.

on behalf of Professor Michael Bruford (Associate Editor) and Kevin Padian (Subject Editor)
openscience@royalsociety.org

Appendix A

Editor comments:

Thank you for your submission. As you will see the reviewers are generally happy with it but have a series of fairly minor concerns that you should be able to address handily. Best wishes in your revisions.

ANSWER: We are grateful for the Editor and Reviewers' positive feedback and valuable comments.

*Numbers inside brackets (e.g. [L.147-150]) refers to line numbers in the clean version of the revised manuscript.

Reviewers' Comments to Author:

Reviewer: 1

Comments to the Author(s)

This is a very good piece of work, contributing additional knowledge on the current distribution of short-eared dogs and assessing future impact of current and projected deforestation rates. The data and methodology are well presented, the results are clear, and the discussion advances the state of knowledge of the distribution of this little known Amazonian canid.

ANSWER: Thank you for the positive feedback and helpful comments.

I found your choice of future horizons as 2027 and 2045 poorly justified by a 6 year generation time which is not supported by any demographic data and clearly too high for a mid-size canid. Also, unclear why to go for a landcover database that is 10 years old, when more recent data is available? Below I list a few comments that I hope will help improve the final version of this work.

The main issue raised by Reviewer1 is our choice of generation time length (6 years). Reviewer1, argues that we should have used a 4-year generation time as proposed by the IUCN Red List assessment for the species (Leite Pitman and Williams, 2011). We based our estimate on the Brazilian Red List assessment for the short-eared dog, which was published two years later by the same first author (Leite Pitman and Beisiegel, 2013). We agree with the Reviewer1 that 6 year might be a long generation time for a canid of this size, however, we do not have any biological information from the species based on field studies to decide between the two generation time length proposed. In view of this lack of reliable data, we decided to stick with the 6-year generation length. We will address this issue, including our reason to keep the longer generation length, in greater depth later in this review.

The issue about the land cover database was also raised by Reviewer2, who raised some additional questions. Please, check our response to Reviewer2 further down on this document (Reviewer2 methods comments).

L106. 'Interfluve' technical term that is not widely used outside technical literature. You should define on first mention

ANSWER: We have added "(i.e portions of land separated by large rivers)" the first time we mention "interfluves" in the manuscript. [L.21]

L110. You can make a recommendation for a Red List reassessment, but should not make the recommendation to what category, since that will only be determined by challenging the RL criteria with your new data. And this paper does not formally do that. You may want to want to say 'upgrading... its Red List status...'

ANSWER: We changed the wording; it now reads "Hence, we propose re-assessing the current short-eared dog's IUCN Red List status of Near Threatened, as upgrading its status may be warranted." [L.26-27]

L130. It's odd to use work that is 20 and 10 years old to support expectations of climate change impact on Amazonian biodiversity. Some of those authors' predictions may be already supported by data, or refuted. There must be more recent work you could consider.

ANSWER: We have added the citation Gomes, Vitor HF et al. "Amazonian tree species threatened by deforestation and climate change." *Nature Climate Change* 9.7 (2019): 547, that corroborates the finding of the classic studies previously cited. [L.46]

L126. and References henceforth. Your reference manager seems to have mangled many authors' names, sometimes including given names, or initials, or listing way too many authors. (eg L146 DEL Maria Renata Pereira. Surname is Leite Pitman; L151. DEL M.R.P.). Please be careful to review your citations carefully before resubmission.

ANSWER: There was a problem with reference manager. We have fixed all citations.

L140. You are using a field guide as a reference for patchy distribution. There are more detailed references (eg IUCN Action Plan; Macdonald & Sillero canid book, Wallace work) to support this statement.

ANSWER: We have made changes as suggested by the reviewer. We have replaced the citation by "Sillero-Zubiri, Claudio, Michael Hoffmann, and David Whyte Macdonald, eds. *Canids: foxes, wolves, jackals, and dogs: status survey and conservation action plan*. Gland, Switzerland: IUCN, 2004." [L.59]

L170 scale-dependent, use hyphen

ANSWER: We have made changes as suggested by the reviewer. [L.103]

L194. Fine to use crab-eating foxes as proxy for HR, but you should look more broadly and cite other studies, not just Courtenay's work. How about hoary foxes?

ANSWER: We have not made changes. The idea was to reduce spatial bias in the locations of the short-eared dog presence records due to unbalanced survey effort across our study region. To do so, we used a study on crab-eating fox home range in the Amazon as a reference to set a minimal distance between our records. Small differences in that distance will have little effect on the final model outputs, as long as we select a reasonable distance. To our knowledge, the Macdonald and Courtenay (1996) study has the most relevant data (closest taxonomic and geographic) we could use to support this decision.

L214 & 216. There should be at least an overlap between datasets. You are using 1970-2000 for

climate, but 2009 for land cover. Why not use more recent climate data, since it is readily available? And why 2006 land cover and not a more recent dataset?

ANSWER: For the climate variables, we used the WorldClim Version2, which has average monthly climate data for minimum, mean, and maximum temperature and precipitation for 1970-2000. This is the most used climate dataset (more than 20000 citations in Google Scholar considering all versions) in species distribution models, particularly for those with continental distribution range. We are not aware of any other climate dataset that is available for our entire study area, at such a fine resolution (~1 km). We added the sentence “To our knowledge, more recent climate variables with comparable resolution were not available”. [L.139-140]

There is more flexibility for the land cover variable. The species location dataset we used in the species distribution modeling has records spanning from 2000 to 2018. We chose GlobCover because it is from 2009, the middle point within in the temporal range of the species record dataset, and has a very good resolution (~300 m).

L232. Soares-Filho 2006 predictions. How good were they 13 years later? That will give you an inkling whether his predictions will be any good going forward.

ANSWER: It is beyond the scope of this study to conduct a formal assessment of how good these future deforestation projections are. However, in the figure below, we visually contrast the future deforestation projected by Soares-Filho et al. (2006) for the year 2018 and the accumulated deforestation mapped by PRODES (2019)* for the same year, which is readily available for the Brazilian Amazon (<http://terrabrasilis.dpi.inpe.br/downloads/>). Based on these figures, Soares-Filho et al. projections have performed well in predicting deforestation, at least for Brazil, the country that encompasses most of the short-eared dog’s distribution. We added the following sentence to the manuscript: “When contrasting Amazon basin forest cover modeled by Soares-Filho et al. (2006) against current maps of deforestation in Brazil (PRODES, 2019), these models have performed well in predicting deforestation, at least for Brazil, the country that encompasses most of the short-eared dog’s distribution (see Figure S1).” [L.159-160]

* The PRODES project performs satellite monitoring and produces annual deforestation rates for the Brazilian Amazon, information that has been used by the Brazilian government since 1988.

Figure 1. Map contrasting future deforestation projected by Soares-Filho et al. (2006) for the year 2018 and the accumulated deforestation mapped by PRODES (2019) for the same for Brazil. This figure is also in the updated supporting material as Figure S1.

L237. Red List criteria. You need to use v3.1 2nd edition (IUCN 2012, NOT IUCN 2001)

ANSWER: We have made changes as suggested by the reviewer. [L.170]

L238. Generation time cannot be 6 years, when most canids of similar or larger since have generation times of 2-3 years (The RL assessment used 4 years). The later will take you to the 10 year window to assess decline (or 12 years, if you use 4 years, but not sure that can be sustained, since there is not actual demographic data to support it). This will work for your 2027 scenario, if you start from the end of your database assemblage. Less clear why you also provide a 2045 scenario. I'd use 30 or 50 years for the long-term scenario, as a matter of personal preference. Justify why use 2045. For a RL reassessment you can look at past changes, or use inferences of future changes.

ANSWER: As mentioned in the manuscript, the short-eared dog is one of the least studied canids in the world. Consequently, there is a lack of information on the short-eared dog's biology, particularly on its demography (e.g. lifespan, reproductive system, sexual maturity, number of broods per year, brood size). To our knowledge, the species does not exist in a captive population in any Zoo or other facilities for ex-situ studies that could be used as reference.

The IUCN (2012) defines generation length as "...the average age of parents of the current cohort (i.e. newborn individuals in the population). ...Generation length is greater than the age at first breeding and less than the age of the oldest breeding individual..." (p.11). Leite Pitman and Williams (2011) propose 4-year generation in the IUCN Red List assessment. Two years later, Leite Pitman and Beisiegel (Leite Pitman and Beisiegel, 2013) updated the generation length to 6 years, neither publication provided supporting biological data, likely because such information is not available. As pointed out by the

reviewer, other canid species of body size similar to the short-eared dog have assumed generation length of 3-4 years (see, IUCN Red List assessment for species of the South American *Lycalopex* genus). Leite Pitman conducted an unpublished study with a tamed male short-eared dog and observed that this individual did not display sexual features (e.g. testicles descended and complex call) until three years of age, which is late for a dog/fox (Hance, 2014). For instance, one-year old male hoary foxes (*Lycalopex vetulus*) are active reproductively (Lemos, F.G., personal communication).

From the conservation perspective, adopting a longer generation length would be a conservative approach, because opting for a shorter generation length implicates that individuals are successful in reproducing earlier in life. In face of the lack of sound data on the short-eared dog reproductive biology and the conservation implication for the species, we decided to keep the 6-year generation length for predicting the species' future distribution. We have added a short version of these arguments to the Methods section to justify our choice [L.171-184]. We also added a possible caveat of our choice in the Discussion section: "These estimates are based on a 6-year generation length, and there is considerable uncertainty about that quantity due to lack of biological data. If the generation length used here is overestimated, our predictions of habitat loss and degradation within 3 generations may be somewhat pessimistic." [L.441-444]

If deemed important by the editor, we can project species distributions for future horizons compatible with 4-year generation length; however this would cause delays the revision process, as we would need to go back to the first stages of the modeling procedure.

L254-257. Explanation a bit woolly. Tighten up.

ANSWER: We made a small change removing unnecessary parts. It now reads "Even though our presence-only species data were not collected using systematic surveys, we believe we meet this assumption, given that covariate values at species record locations cover most of the distribution of covariates across our area of inference (see Figure S3)." [L.199-202]

L286. Finally, I found a definition of interfluvial. Should be presented on first mention.

ANSWER: We have made changes as suggested by the reviewer. [L.21]

L287. Definition unclear. Are you saying that protected areas INCLUDE indigenous land? This will require different wording.

ANSWER: To make it clear that we are including indigenous land as protected areas, we have changed the sentence. It now reads "We assessed the current and future total area predicted to be occupied by the short-eared dog per country (including all countries in which the species has been recorded), per interfluvial (Amazon basin sub-regions limited by the main rivers, see Figure S4) and within/outside protected areas and indigenous land (hereafter collectively referred to as protected areas)." [L.229-232] We are open to change the word if deemed necessary by the editor.

L304. This methodology needs greater detail. It does not tell me how many of those cameras were deployed at a given time. 180 cameras at the same time?, 60 moved to three different arrays at different times? Also how far apart were the cameras set up from each other?

ANSWER: We have added more details on the camera trapping surveys. It now reads “We compiled data from four camera trap surveys, adding up to 180 sites surveyed within the study area between February 2016 and June 2017.” [L.249-250] We also added the information on spacing between camera trap site: “Camera traps operated 24 hours/day, with no delay between subsequent potential triggers, and in average 3640 m (range = 815 – 40547 m) apart from neighboring camera trap sites.” [L.257-258]

L327-330. The binary forest layer, where does it come from? Or you produced it yourselves?

ANSWER: The Global Forest Cover layer (GFC, Hansen et al., 2013) represents estimates of the percentage of a pixel covered by woody vegetation greater than 5 meters in height and is available online. Based on Tan et al. (2017), we used the 75% threshold (pixels $\geq 75\%$ were considered forested) to create a binary forest cover layer, which were then used to calculate both fragmentation metrics. We have added the Tan et al. (2017) reference, where this procedure is described in detail. [L.276-277]

L347. Have you defined camera trap effort somewhere?

ANSWER: We have added a definition of camera trap effort here. It now reads “We considered camera-trap effort (i.e. number of days camera traps were active during each 20-day sampling occasion) as a survey covariate.” [L.281-282]

L368. This has to be predicted for 10 years (or 12, if you can justify a 4 year generation time). You should not use 6x3, since it is not biologically sound.

ANSWER: Please see above comments from Reviewer1 for the line L238.

L378 change ‘any’ to ‘either’

ANSWER: We have made changes as suggested by the reviewer.

L416. This excluded the most recent RL assessment Leite & William 2011

ANSWER: Please see above comments from Reviewer1 for the line L238.

L423. Not sure I understand why ‘Most’ recent, and not all? That happened to the exceptions? Explain.

ANSWER: The reviewer is correct. All, and not most, of the new records fall within our proposed distribution. We have made changes as suggested by the reviewer. [L.372]

L485. I have an issue with the choice of time window. When the RL reassessment takes place, it will be based on a 10 year window (with a realistic generation time of 2-3 years). Therefore, for this paper to be informative you should provide prediction for the right time interval.

ANSWER: Please see above comments from Reviewer1 for the line L238.

L510. You need to properly reference your mention of disease as a potential risk.

ANSWER: We have added Lainson et al. (1994) to back up our claim that the increased expansion of human occupation in the Amazon might contribute to increased diseases risk for wild species. [L.466]

L513. No need to repeat scientific name.

ANSWER: We have deleted the scientific name in this sentence.

L516-517. This is a worthwhile recommendation but it should be a recommendation to reassess the species status, in light of the new data and analysis. You can't pre-adjudicate this assessment will necessarily be to 'Vulnerable'. That will be the outcome of testing against the criteria. It is, after all, a quantitative process, not a qualitative one. Reword accordingly.

ANSWER: We have made changes to as suggested by the reviewer. It now reads "Considering the magnitude of the predicted negative effects of forest loss on the short-eared dog, we suggest re-assessment of the species' current IUCN status (Near Threatened) throughout its distribution, as an upgrade of its threat category seems warranted." [L.471-473]

L574. Address lots of typos and formatting.

ANSWER: We have reviewed the manuscript thoroughly.

Reviewer: 2

Comments to the Author(s)

Wild dogs at stake: Deforestation threatens the only Amazon endemic canid, the short-eared dog (*Atelocynus microtis*); Rocha et al.; RSOS

Rocha et al. present a fascinating study of the severely under-studied short-eared dogs, combining historic presence records, camera trapping and species distribution models to predict current and future distribution of the species. The authors also present a multi-scale perspective (regional and local), which sheds some light on the scale-specific factors affecting the species' association with habitat. Overall, the paper is very well written and the authors have done a commendable job—considering the effort it would have taken to coordinate, initiate collaborations and compile information about a rare species from multiple sources and institutions. I do have some concerns related to the local scale analysis (explained below).

ANSWER: Thank you for the positive feedback. We address the issues raised by the reviewer about the local scale analysis below.

Introduction:

L127: Please revise the sentence: "Besides the alarming deforestation rates, there is growing concern..."

ANSWER: We have made changes as suggested by the reviewer. [L.47]

Methods:

General comments: If I understood correctly, the authors use all presence records (post-rarifying) from 2000 onwards. Does this include data from 2009 to 2018/19? If so, there would be some mismatch in the forest cover (i.e. potential habitat) and current distribution right? The forest cover layer is from 2009, but presence records are from subsequent years as well. I get that this may be a limitation because of access to updated forest cover data. But this issue needs to be acknowledged in the Methods section.

ANSWER: It is possible that some of the species presence records post-2009 are at sites that experienced vegetation cover change during 2009-2018.

The PRODES project makes spatial information of yearly-deforested areas in the Brazilian Amazon readily available (<http://terrabrasilis.dpi.inpe.br/downloads/>). To take a closer look at this possible temporal vegetation cover mismatch, we checked how many of our short-eared dog records from Brazil fall in areas deforested after 2009. Out of 271 post-2000 short-eared dog records from Brazil, only 2 records are located at recently deforested site. As most of the records are from Brazil, we believe that the vegetation mismatch is not a big issue across the entire distribution. Additionally, most (>65%) of the post-2009 records, considering all country, are from camera trap surveys within protected areas, where deforestation is less likely to happen. We have added a short version of these arguments to the main text. [L.143-148]

Local scale analysis: Given that that authors list habitat loss, prey decline, disease from domestic dogs and other anthropogenic factors (road related mortality) as factors that negatively impact short-eared dogs, I wonder why none of these covariates, or variables that could serve as surrogates of these factors, were not considered for modeling habitat use. The authors also list lowland forests, areas close to water etc. as purported “good habitats” for the focal species. The large-scale predicted distribution also clearly indicates importance of protected areas for the species to persist in the long term. Elevation, distance to protected area boundary, distance to water bodies, proximity to human settlement, human footprint index, encounters of hunters/logging machinery, encounters of domestic dogs etc. are all really good potential covariates that would make the local scale analysis very interesting while also better informing management/conservation. The three covariates used to model occupancy PSI may be too broad to have show any influence at the local scale; this is evident in the estimates of the slope co-efficients. Both forest cover and edge density have high SE values (95% CI overlaps 0, so no statistical significance), suggesting weak effects of the predictors. I strongly suggest that the authors re-analyse the local-scale data incorporating some/all the covariates listed above.

ANSWER: We did not included different site variables (e.g. distance to water, roads) as predictors of short-eared dog because we were originally interested in understanding how forest composition and configuration affect the species’ occupancy (local scale), as forest cover was the main predictor for the species’ distribution in the species distribution modeling (regional scale). However, we ran some additional analyses to assess the effects of some of the variables suggested by the Reviewer2. From the list of potential site covariates compiled by the reviewer, we could acquire reliable spatial layers for four: distance to water (from Seyler et al., 2009), distance to roads (from RAISG, available at <https://www.amazoniasocioambiental.org/en/maps/#download>), elevation (from Jarvis et al., 2008) and Human Footprint Index (from Venter et al., 2016). We do not have sufficient records of domestic dogs, hunters, logging machinery to include them as a covariate of human disturbance. We were also not able to reliably estimate difference in species occurrence in protected vs unprotected areas because only one short-eared dog record was outside protected areas in our local scale study.

We followed the same procedures described in the manuscript to generate and select covariates (including the scale-optimization procedure) and ran occupancy models including the four new variables. However, none of the four covariates performed better than the baseline model {p(EFF) psi(.)}

(see Table 1 below), regardless of the buffer size, indicating that those variable had no detectable effects on species occupancy in our study. We highlight that our study was specifically designed to assess effects of forest configuration and therefore, we cannot make strong inference on other potential drivers of the species' space use.. Consequently, we have not added this new analysis to the manuscript, but we briefly mention this in the methods [L.277-280] and discussion [L.456-459] sections and direct interested readers to check the supplemental materials, which we updated with the table below.

Table 1. Results of AIC-based model selection for occupancy models for short-eared dog (*Atelocynus microtis*) in the Southern Brazilian Amazon. Single-covariate single-season occupancy models were compared to select the best buffer radius around each camera-trap for each covariate. The site covariates tested were distance to water (Water), distance to roads (Road) elevation (Elev) and Human Footprint Index) (HFI) in different buffer radius (0.5, 1, 1.5, 3, 5, 10 km). Effort (EFF) was a survey covariate included in all models. Δ AIC is the relative difference in AIC values compared with the top ranked model.

Model	Parameters	AIC	delta AIC
Distance to water			
p(EFF) psi(.)	3	186.2	0
p(EFF) psi(Water_b5)	4	187.5	1.25
p(EFF) psi(Water_b10)	4	187.6	1.36
p(EFF) psi(Water_b3)	4	187.8	1.58
p(EFF) psi(Water_b1.5)	4	188.2	1.92
p(EFF) psi(Water_b1)	4	188.2	1.96
p(EFF) psi(Water_b0.5)	4	188.2	1.99
Distance to roads			
p(EFF) psi(.)	3	186.2	0
p(EFF) psi(Road_b5)	4	195.4	9.16
p(EFF) psi(Road_b3)	4	195.5	9.3
p(EFF) psi(Road_b1.5)	4	196.5	10.3
p(EFF) psi(Road_b1)	4	196.7	10.41
p(EFF) psi(Road_b0.5)	4	196.7	10.45
p(EFF) psi(Road_b10)	4	229.9	43.65
Elevation			
p(EFF) psi(.)	3	186.2	0
p(EFF) psi(Elev_b1.5)	4	187.7	1.41
p(EFF) psi(Elev_b3)	4	187.7	1.42
p(EFF) psi(Elev_b1)	4	187.8	1.61
p(EFF) psi(Elev_b0.5)	4	187.9	1.71
p(EFF) psi(Elev_b5)	4	188.0	1.8
p(EFF) psi(Elev_b10)	4	188.2	1.95
Human Footprint Index			
p(EFF) psi(.)	3	186.2	0
p(EFF) psi(HFI_b0.5)	4	186.5	0.28
p(EFF) psi(HFI_b1)	4	187.0	0.74
p(EFF) psi(HFI_b1.5)	4	187.4	1.14

p(EFF) psi(HFI_b3)	4	187.9	1.61
p(EFF) psi(HFI_b10)	4	187.9	1.63
p(EFF) psi(HFI_b5)	4	188.0	1.77

L335: revise the sentence to state that the sample units are “likely” smaller than the “presumed” home-range size of the species.

ANSWER: We have made changes as suggested by the reviewer. [L.286]

Results:

General comments: Move Table S4 to main manuscript. This is the primary result of your study. Similarly, combine three panels of Figures S5, S6, S7 and present it as one figure in main manuscript. The results related to local-scale analysis would change when the data are re-analysed. Please present revised results accordingly.

ANSWER: We have made changes suggested by the reviewer. Table S4 is now in the main text as Table 1 [L.738-743]. Figure S5-S7 were combined and are in the main text as Figure 6 [L.792-797]. The new variables suggested by the reviewer that were included in the local scale (occupancy) analysis did not lead to changes in the variables and model selection, and therefore our local scale results remained unaltered.

Discussion:

General comments: The section is well written for the most part, and makes a compelling case for revising the IUCN status of short eared dog. I think the paper would benefit from expanding the narrative, both, in the Introduction and in the Discussion sections. As it stands, the focus is a bit narrow. If at all the species does qualify to be considered a “flagship” for conservation of forest systems in the Amazon, then the entire Discussion should make a more convincing case by establishing links between the short-eared dogs, their prey, competing predators and humans. The authors go some length in doing so in the last paragraph. But the argument is lacking in that it only makes a passing mention of the benefit to other species.

ANSWER: We have made considerable changes in the entire introduction section to make it broader and more appealing to a diverse readership. Instead of focusing on the Amazon and on our target species, the updated introduction starts considering the importance of understanding the drivers of species’ distribution and the need of multi-scale studies. In the last paragraph of the introduction we argue why the short-eared dog is a good model species, how it relates to other understudied and vulnerable species, as well as how our analytical approach can be transferable to other studies [L.36-51 and 87-98]. In the end of discussion, we refer back to those topics mentioned in the introduction and discuss how our approach, that integrates data across sources and modeling procedures, can be used to improve our knowledge on understudied species. We also discuss how other mammals share vulnerabilities with the short-eared dog in the Amazon, as well as in other ecosystems [L.478-493].

References

- Jarvis, A., Reuter, H.I., Nelson, A., Guevara, E., 2008. Hole-filled SRTM for the globe Version 4. available from CGIAR-CSI SRTM 90m Database (<http://srtm.csi.cgiar.org>) 15.
- Leite Pitman, M.R.P., Williams, R.S.R., 2011. *Atelocynus microtis* [WWW Document]. IUCN Red List Threat. Species 2011 e.T6924A12814890. URL <http://dx.doi.org/10.2305/IUCN.UK.2011-2.RLTS.T6924A12814890.en>. (accessed 4.22.16).
- Leite Pitman, R., Beisiegel, B. de M., 2013. Avaliação do risco de extinção do cachorro-do-mato-de-orelhas-curtas *Atelocynus microtis* (Sclater, 1883) no Brasil. *Biodiversidade Bras.* 3, 133–137.
- PRODES, 2019. Projeto PRODES: monitoramento da floresta amazônica brasileira por satélite [WWW Document]. URL <http://www.obt.inpe.br/prodes/index.php>
- Seyler, F., Muller, F., Cochonneau, G., Guimarães, L., Guyot, J.L., 2009. Watershed delineation for the Amazon sub-basin system using GTOPO30 DEM and a drainage network extracted from JERS SAR images. *Hydrol. Process. An Int. J.* 23, 3173–3185.
- Soares-Filho, B.S., Nepstad, D.C., Curran, L.M., Cerqueira, G.C., Garcia, R.A., Ramos, C.A., Voll, E., McDonald, A., Lefebvre, P., Schlesinger, P., 2006. Modelling conservation in the Amazon basin. *Nature* 440, 520–523.
- Tan, C.K.W., Rocha, D.G., Clements, G.R., Brenes-Mora, E., Hedges, L., Kawanishi, K., Mohamad, S.W., Mark Rayan, D., Bolongon, G., Moore, J., Wadey, J., Campos-Arceiz, A., Macdonald, D.W., 2017. Habitat use and predicted range for the mainland clouded leopard *Neofelis nebulosa* in Peninsular Malaysia. *Biol. Conserv.* 206, 65–74. doi:10.1016/j.biocon.2016.12.012
- Venter, O., Sanderson, E.W., Magrath, A., Allan, J.R., Beher, J., Jones, K.R., Possingham, H.P., Laurance, W.F., Wood, P., Fekete, B.M., 2016. Global terrestrial Human Footprint maps for 1993 and 2009. *Sci. data* 3, 160067.

Appendix B

Wild dogs at stake: Deforestation threatens the only Amazon endemic canid, the short-eared dog (*Atelocynus microtis*); Rocha et al.; RSOS; R1

I appreciate the authors' detailed responses and justifications to my previous comments. They have also done a good job with revising the narrative based on suggestions made by the reviewers.

One of my primary concerns in the previous version was with the local-scale analysis. I see that the authors have used some additional covariates based on my suggestions and analysed the data again (presented in Table S8). I must apologize for my oversight here, but I missed seeing this error in the previous version. The models where the authors have used $p(\text{EFF})$, $\psi(\cdot)$ should have THREE parameters (p , beta-effort, ψ), and associated AIC would be 186.2 (as is the case for all the model sets with the new covariates). But for forest cover, edge density and patch density, the same model seems to have only TWO parameters and the AIC is 230.9. There is an error either in the new set of covariate models (water, road, elevation HFI), or in the forest-based covariates from before. If the correct AIC for $p(\text{EFF})$, $\psi(\cdot)$ is actually 186.2, then this model would consistently rank the highest (least AIC) across all model combinations, and none of the models even with forest-based covariates would be statistically significant.

If my observation above is correct, then I suggest the authors check for additive and/or interactive effects of the covariates. This may produce some covariate models that rank higher than $p(\text{EFF})$, $\psi(\cdot)$, and the authors could revise their results and discussion based on that. If, by chance, even after these changes the model with least AIC happens to be $p(\text{EFF})$, $\psi(\cdot)$, then the authors can present all the models/results, and deliberate on the “indicative” effects (magnitude and direction) of the covariates, while explicitly stating that none of the covariate models received adequate statistical support.

Minor comments:

L26–29: I suggest rewording this to “We propose a re-assessment of the short-eared dog’s current IUCN Red List status (Near Threatened) based on findings presented here. Our study also exemplifies how data can be integrated across sources and modeling procedures to improve our knowledge of relatively understudied species.”

L40–41: Reword: “...given the recent human-induced environmental changes in Amazonia.”

L49: Replace “predict species distribution” with “predict distribution patterns”

L81–82: Suggested edit: “we used occupancy models to investigate how habitat use by the short-eared dog is affected by attributes related to forest cover at a fine spatial scale (local scale).”

L83–86: Delete these lines, or include them in the Discussion (but I think a slightly different version is already in the Discussion).

Paragraph L87–L98: May be a better fit as the penultimate paragraph of the Discussion section.

L92–93: “large- and small-scale”

L93: Replace “as a forest dweller” to “as a forest-dependent species”

Appendix C

Associate Editor Comments to Author (Professor Michael Bruford):

The two referees who reviewed your manuscript have come back with additional comments, one of which potentially requires additional analysis. Please consider these requests and carry out the additional analysis if necessary

ANSWER: Thank you; please see our response to the reviewer comments below.

Subject Editor Comments to Author:

One reviewer feels strongly that you did not address what appears to be an unrealistic estimate of generation time, given what is known of canids. These and other concerns will need to be addressed fully in your final version of submission; please be explicit and best wishes for your revisions.

ANSWER: Thank you; please see our response to the reviewer comments below.

Reviewer comments to Author:

Reviewer: 1

Comments to the Author(s)

As stated in my original review this is a very good piece of work, contributing additional knowledge on the current distribution of short-eared dogs and assessing future impact of current and projected deforestation rates. The data and methodology are well presented, the results are clear, and the discussion advances the state of knowledge of the distribution of this little known Amazonian canid. That assessment still stands. However, I was disappointed by the authors' reluctance to review the value of their guessed generation time for the species (not supported by any demographic data). A 6 year generation time does not reflect what we know about the reproductive biology of similar sized canids. Therefore a proxy generation time should be used, revised down to 3 years (max 3.5 years) to reflect that observed across similar size canids. Future horizons should change accordingly. The choice of such a conservative generational time weakens the authors' argument for uplisting the species Red List status.

ANSWER: We feel that this question is not one of right or wrong (as there are essentially no data on reproductive ecology of this species, from the wild or captivity), but rather of opinion – whether to use the suggested generation length published in 2011 or that published in 2013. We opted for the latter, because, as we already laid out in the previous response to reviewer comments, it is the more recent value, and both were published by the same first author (who is also a coauthor of this study). In the last round of revisions, we added justification and caveats for this choice to the manuscript [L.171-184 and L.441-444 of the clean version of our manuscript].

To further evaluate potential consequences of choosing one generation length over the other for our analysis, we looked at the amount of deforestation projected to happen within the short-eared dog's distribution over three times each generation length (according to IUCN criterion of loss of distribution over 3 generations). Based on the most likely scenario (Business as Usual), the proportion of the current short-eared dog distribution deforested by 2027 (6-year generation length) is estimated to be 24.9%. If we adopt Reviewer 1's suggestion of a 4-year generation length (i.e., if we look at projected deforestation by year 2021), this proportion changes to 21%. As in our model the changes in predicted

short-eared dog distribution are driven by changes in forest cover, this small difference in deforestation strongly suggests that changing the generation length will not have strong impacts on our projections of future short-eared dog distribution and thus, the conclusions of our study, but will require substantial extra work. We included a statement to this effect in line **L.444-449**.

Reviewer: 2

Wild dogs at stake: Deforestation threatens the only Amazon endemic canid, the short-eared dog (*Atelocynus microtis*); Rocha et al.; RSOS; R1

I appreciate the authors' detailed responses and justifications to my previous comments. They have also done a good job with revising the narrative based on suggestions made by the reviewers.

ANSWER: We are grateful for the positive feedback.

One of my primary concerns in the previous version was with the local-scale analysis. I see that the authors have used some additional covariates based on my suggestions and analysed the data again (presented in Table S8). I must apologize for my oversight here, but I missed seeing this error in the previous version. The models where the authors have used $p(\text{EFF})$, $\psi(\cdot)$ should have THREE parameters (p , beta-effort, ψ), and associated AIC would be 186.2 (as is the case for all the model sets with the new covariates). But for forest cover, edge density and patch density, the same model seems to have only TWO parameters and the AIC is 230.9. There is an error either in the new set of covariate models (water, road, elevation HFI), or in the forest-based covariates from before. If the correct AIC for $p(\text{EFF})$, $\psi(\cdot)$ is actually 186.2, then this model would consistently rank the highest (least AIC) across all model combinations, and none of the models even with forest-based covariates would be statistically significant.

If my observation above is correct, then I suggest the authors check for additive and/or interactive effects of the covariates. This may produce some covariate models that rank higher than $p(\text{EFF})$, $\psi(\cdot)$, and the authors could revise their results and discussion based on that. If, by chance, even after these changes the model with least AIC happens to be $p(\text{EFF})$, $\psi(\cdot)$, then the authors can present all the models/results, and deliberate on the "indicative" effects (magnitude and direction) of the covariates, while explicitly stating that none of the covariate models received adequate statistical support.

ANSWER: Thank you for spotting this error in the table. There is no error in our analysis or conclusion, this is just a mistake in the table. The first version of this manuscript submitted to RSOS had only forest cover, edge density and patch density as occupancy variables. We later added proximity to water and roads, elevation and human footprint index following Reviewer 2's suggestion. As none of these additional variables had a significant effect, I just added these results to the table and did not rerun the models for the three original variables. Now that Reviewer 2 pointed out the mismatch in the AIC values, we reran the models for all covariates and I updated the tables accordingly in the manuscript and supporting material. Some AIC values have changed, but the model rank, and thus our conclusions, did not change at all. For the sake of transparency, we added a print screen of the R model list outputs. The AIC values for the models for the three original occupancy variables should not have changed between runs, as they use the exactly same input data and R script. Most likely, some error occurred copying R output to the original manuscript table.

```
> modSel(modlist.forest)
```

	nPars	AIC	delta	AICwt	cumltvWt
mod_p.eff_psi.Fl6b0.5	4	178.73	0.00	0.519	0.52
mod_p.eff_psi.Fl6b1	4	179.93	1.20	0.285	0.80
mod_p.eff_psi.Fl6b1.5	4	182.41	3.68	0.083	0.89
mod_p.eff_psi.Fl6b3	4	183.34	4.61	0.052	0.94
mod_p.eff_psi.Fl6b5	4	185.05	6.32	0.022	0.96
mod_p.eff_psi.Fl6b0.1	4	185.57	6.84	0.017	0.98
mod_p.eff_psi.	3	186.23	7.50	0.012	0.99
mod_p.eff_psi.Fl6b10	4	186.53	7.80	0.011	1.00

```
> modSel(modlist.ED)
```

	nPars	AIC	delta	AICwt	cumltvWt
mod_p.eff_psi.EDb1	4	184.85	0.00	0.250	0.25
mod_p.eff_psi.EDb3	4	185.47	0.62	0.184	0.43
mod_p.eff_psi.EDb1.5	4	185.62	0.77	0.170	0.60
mod_p.eff_psi.EDb10	4	186.23	1.38	0.126	0.73
mod_p.eff_psi.	3	186.23	1.38	0.125	0.86
mod_p.eff_psi.EDb5	4	186.71	1.86	0.099	0.95
mod_p.eff_psi.EDb0.5	4	188.23	3.38	0.046	1.00

```
> modSel(modlist.PD)
```

	nPars	AIC	delta	AICwt	cumltvWt
mod_p.eff_psi.	3	186.23	0.00	0.248	0.25
mod_p.eff_psi.PDb1.5	4	186.81	0.58	0.186	0.43
mod_p.eff_psi.PDb1	4	187.23	0.99	0.151	0.58
mod_p.eff_psi.PDb3	4	187.68	1.44	0.120	0.70
mod_p.eff_psi.PDb10	4	187.86	1.63	0.110	0.81
mod_p.eff_psi.PDb0.5	4	188.19	1.95	0.093	0.91
mod_p.eff_psi.PDb5	4	188.20	1.97	0.093	1.00

```
> modSel(modlist.water)
```

	nPars	AIC	delta	AICwt	cumltvWt
mod_p.eff_psi.	3	186.23	0.00	0.28	0.28
mod_p.eff_psi.waterb5	4	187.49	1.25	0.15	0.42
mod_p.eff_psi.waterb10	4	187.60	1.36	0.14	0.56
mod_p.eff_psi.waterb3	4	187.82	1.58	0.13	0.69
mod_p.eff_psi.waterb1.5	4	188.15	1.92	0.11	0.79
mod_p.eff_psi.waterb1	4	188.19	1.96	0.10	0.90
mod_p.eff_psi.waterb0.5	4	188.23	1.99	0.10	1.00

```
> modSel(modlist.road)
```

	nPars	AIC	delta	AICwt	cumltvWt
mod_p.eff_psi.	3	186.23	0.00	9.6e-01	0.96
mod_p.eff_psi.roadb5	4	195.40	9.16	9.9e-03	0.97
mod_p.eff_psi.roadb3	4	195.54	9.30	9.2e-03	0.98
mod_p.eff_psi.roadb1.5	4	196.54	10.30	5.6e-03	0.99
mod_p.eff_psi.roadb1	4	196.65	10.41	5.3e-03	0.99
mod_p.eff_psi.roadb0.5	4	196.68	10.45	5.2e-03	1.00
mod_p.eff_psi.roadb10	4	229.89	43.65	3.2e-10	1.00

```

> modSel(modlist.elev)
      nPars      AIC delta AICwt cumltvWt
mod_p.eff_psi.      3 186.23  0.00  0.27    0.27
mod_p.eff_psi.elevb1.5  4 187.65  1.41  0.14    0.41
mod_p.eff_psi.elevb3    4 187.65  1.42  0.14    0.55
mod_p.eff_psi.elevb1    4 187.84  1.61  0.12    0.67
mod_p.eff_psi.elevb0.5  4 187.94  1.71  0.12    0.78

```

Minor comments:

L26–29: I suggest rewording this to “We propose a re-assessment of the short-eared dog’s current IUCN Red List status (Near Threatened) based on findings presented here. Our study also exemplifies how data can be integrated across sources and modeling procedures to improve our knowledge of relatively understudied species.”

ANSWER: We have made changes as suggested by the reviewer. [L.26-29]

L40–41: Reword: “...given the recent human-induced environmental changes in Amazonia.”

ANSWER: We have made changes as suggested by the reviewer. [L.40-41]

L49: Replace “predict species distribution” with “predict distribution patterns”

ANSWER: We have made changes as suggested by the reviewer. [L.49]

L81–82: Suggested edit: “we used occupancy models to investigate how habitat use by the short-eared dog is affected by attributes related to forest cover at a fine spatial scale (local scale).”

ANSWER: We have made changes as suggested by the reviewer. [L.81-83]

L83–86: Delete these lines, or include them in the Discussion (but I think a slightly different version is already in the Discussion).

ANSWER: We have not made changes here. This study has several outputs and those lines describe those outputs in a logical way, preparing the reader to what lays ahead in the paper. Therefore, we see value in keeping those lines. [L.83-86]

Paragraph L87–L98: May be a better fit as the penultimate paragraph of the Discussion section.

ANSWER: This paragraph was not in the manuscript we submitted to RSOS and was added during the revision following the suggestion to provide a short paragraph in the introduction section on the broader impact of your study.

L92–93: “large- and small-scale”

ANSWER: We have made changes as suggested by the reviewer. [L.92-93]

L93: Replace “as a forest dweller” to “as a forest-dependent species”

ANSWER: We have made changes as suggested by the reviewer. [L.93]

Appendix D

Associate Editor Comments to Author (Professor Michael Bruford):

Comments to the Author:

Thanks for your revision which I think places the MS close to acceptance. I understand your stance on the generation time estimate and I agree with you that this is justified. I would just like to see you address it a bit more in the discussion perhaps with a caveat that using 6 years might change the scenario somewhat, for transparency.

ANSWER: In this study, we present several arguments, based on proportions of the species' distribution area expected to be affected by deforestation, to suggest a re-assessment of the short-eared dog's IUCN category. This is the ultimate consequence of our generation length choice. To address Dr. Bruford's request, we added (right after suggesting the re-assessment) the following sentence in the Discussion section: "Once again, we highlight that our suggestion to re-assess the short-eared dog's IUCN status is based on a 6-year generation length. Shorter generation lengths might result in distribution/habitat loss that do not satisfy IUCN criteria for upgrading the species threat category." [L.485-488] To make it clear how our generation length choice might change the scenario, we also added to the discussion: "Thus, if short-eared dog's generation length is shorter than assumed in this study, our future prediction horizons would also be shorter and a smaller proportion of the species' distribution would be expected to be impacted by deforestation." [L.444-447]